# Robust Decision Making with Partially Calibrated Forecasts

**Shayan Kiyani , Hamed Hassani, George Pappas & Aaron Roth**
University of Pennsylvania
{shayank, hassani, pappasg, aaroth}@seas.upenn.edu

## Abstract

Calibration has emerged as a foundational goal in "trustworthy machine learning", in part because of its strong decision theoretic semantics. Independent of the underlying distribution, and independent of the decision maker's utility function, calibration promises that amongst all policies mapping predictions to actions, the uniformly best policy is the one that "trusts the predictions" and acts as if they were correct. But this is true only of *fully calibrated* forecasts, which are tractable to guarantee only for very low dimensional prediction problems. For higher dimensional prediction problems (e.g. when outcomes are multiclass), weaker forms of calibration have been studied that lack these decision theoretic properties. In this paper we study how a conservative decision maker should map predictions endowed with these weaker ("partial") calibration guarantees to actions, in a way that is robust in a minimax sense: i.e. to maximize their expected utility in the worst case over distributions consistent with the calibration guarantees. We characterize their minimax optimal decision rule via a duality argument, and show that surprisingly, "trusting the predictions and acting accordingly" is recovered in this minimax sense by *decision calibration* (and any strictly stronger notion of calibration), a substantially weaker and more tractable condition than full calibration. For calibration guarantees that fall short of decision calibration, the minimax optimal decision rule is still efficiently computable, and we provide an empirical evaluation of a natural one that applies to any regression model solved to optimize squared error.

## 1 Introduction

Machine learning systems are increasingly deployed in high-stakes decision making domains such as healthcare, finance, and law. The predictive power of these models can be extraordinary, but scoring well on predictive error metrics does not directly guarantee that decisions downstream of those predictions will be correct. For predictions to be operationally useful, a decision-maker must be able to treat them as reliable inputs into a downstream decision making policy. This raises two fundamental questions:

**On the Model Side:** *What does it mean for machine learning predictions to be trustworthy in decision-making contexts?*

**On the Decision Making Side:** *Given predictions that satisfy a particular type of "trustworthiness", how should the decision maker adapt its actions to the promised guarantees?*

**On the Model Side:** A natural answer is that trustworthy predictions should directly support good decisions as they are. In other words, the decision-maker should be able to reliably best respond to the forecaster's predictions as if they were correct. Formally, let $(X, Y)$ be a pair of random variables drawn from a joint distribution $\mathcal{D}$, where $X \in \mathcal{X}$ represents the observed features and $Y \in [0, 1]^d$ is the outcome of interest. Let $\mathcal{A}$ denote the action set, and suppose the decision-maker follows a policy $a(\cdot) : [0, 1]^d \to \mathcal{A}$ mapping predictions to actions. Given a predictor $f$, the decision maker's performance when using a policy $a$ is measured by its expected utility on the underlying distribution:

$$\mathbb{E}_{(X,Y) \sim \mathcal{D}}[u(a(f(X)), Y)],$$

where $u(a, y) \in \mathbb{R}$ is a utility function. Given a forecaster $f : \mathcal{X} \to [0, 1]^d$, the *plug-in best response* to a forecast is defined as

$$a_{\mathrm{BR}}(f(x)) \;=\; \arg\max_{a \in \mathcal{A}} \; u(a, f(x)). \tag{1}$$

Thus, a forecaster $f$ is trustworthy if the decision-maker's best-response policy $a_{\mathrm{BR}}(f(x))$ achieves higher utility than any other policy. When is this the case?

The classical answer lies in the notion of *calibration*. Intuitively, a forecaster is calibrated if, whenever it predicts a vector $f(x) = v \in [0, 1]^d$, the empirical outcomes are consistent with that prediction. More formally, a forecaster $f$ is said to be *fully calibrated* if for every $v \in [0, 1]^d$,

$$\mathbb{E}[Y \mid f(X) = v] = v.$$

It is well known that best responding to calibrated forecasts is the optimal decision policy among all policies that map forecasts to actions (Foster & Vohra, 1997; Kleinberg et al., 2023; Noarov et al., 2023; Roth, 2022).

However, achieving full calibration is extremely difficult, both in theory—the sample complexity of calibrating an existing forecaster without harming its accuracy grows exponentially with the outcome dimension $d$ (Gopalan et al., 2024a)—and in practice, where empirical evidence shows systematic deviations from calibration, ranging from neural networks to large language models (Guo et al., 2017; Kull et al., 2019; Gupta & Ramdas, 2022; Plaut et al., 2024). Thus, despite the appealing link between calibration and trustworthy ML-powered decision-making, this connection quickly breaks down in real-world applications.

**On the Decision Making Side:** Decision making from predictions admits two canonical extremes. At one end, the decision maker *aggressively best responds* to the forecasts, acting as if they were fully correct. At the other end, the decision maker *conservatively plays a minimax-safety strategy*, $\arg\max_{a \in \mathcal{A}} \min_{y \in \mathcal{Y}} u(a, y)$, treating the forecasts as if they carried no information about the instance.

Departing from these extremes, we treat a model $f$ and its forecast $f(x)$ as information that constrains what the true, instance-conditional outcome distribution could be. In other words, after observing $f(x)$, the decision maker considers the set of *candidate realities*—outcome distributions consistent with the forecast and the available calibration guarantees. Intuitively, the "volume" of this set is governed by the strength of calibration: under full calibration, the set collapses to the forecast itself (the prediction can be treated as reality, at least in expectation), whereas as calibration weakens, the set enlarges. A principled decision rule should therefore *tune its conservatism to what the reality could be*, consistent with the provided guarantees. This idea, together with the fragility of full calibration in practice, leads to the central question of this paper: *can we derive optimal decision-making policies under weaker and more practical conditions than full calibration?*

We answer this question affirmatively. We introduce a framework based on *conservative* decision making that nevertheless fully exploits *partially* calibrated forecasts. This viewpoint echoes ideas in robust optimization and control, but it has not been systematically developed for post hoc decision making with partially calibrated machine-learning forecasts.

## 1.1 Our Results

We consider a parameterized family of weighted calibration guarantees that have recently become a popular object of study (Hébert-Johnson et al., 2018; Gopalan et al., 2022). Informally speaking, this family of guarantees constrains the residuals of a predictor $f$ to be uncorrelated with a collection of "test functions" $h \in \mathcal{H}$ mapping the range of $f$ to the reals. When $\mathcal{H}$ consists of all such test functions, we recover full calibration, but many popular variants of calibration (e.g. top label calibration, decision calibration, etc) can be expressed as instances of $\mathcal{H}$-calibration under much smaller/more tractable sets $\mathcal{H}$. Our contributions are as follows:

1. In Section 2 we formalize the following question: given a set of test functions $\mathcal{H}$ and a predictor $f(x)$ that is promised to satisfy $\mathcal{H}$-calibration, what decision rule $a : [0, 1]^d \to \mathcal{A}$, mapping predictions to actions, will maximize a decision maker's expected utility in the worst case over all joint distributions over $X \times Y$ that are consistent with the promise that $f$ is $\mathcal{H}$-calibrated?

2. In Section 3 we answer this question by giving a closed-form for the decision maker's optimal decision rule, in terms of the dual variables of a convex program that can be efficiently computed for any finite $\mathcal{H}$.

3. In Section 4 we instantiate this decision rule for various calibration guarantees of interest. Of particular note, we find that when $\mathcal{H}$ corresponds to the tractable notion of *decision calibration* (Zhao et al., 2021; Noarov et al., 2023), then the optimal decision rule is the best response decision rule $a_{\mathrm{BR}}$, just as it is for (the intractable notion of) full calibration. In fact, it suffices that $\mathcal{H}$ *contains* the decision calibration constraints — any larger set *also* makes best response the optimal decision rule. Thus what could have been a very large hierarchy of minimax optimal decision rules "collapses" to best response at the level of decision calibration. An upshot of this is that a predictor can be simultaneously decision calibrated for many downstream decision makers, and for each of them, best response will be their optimal decision policy in this minimax sense. We also derive the minimax optimal decision rule for a simple "self-orthogonality" calibration condition that will hold for any regression model with a linear final layer trained to optimize squared loss, and hence will be commonly satisfied without any algorithmic intervention.

4. In Section 5 we train a two-layer MLP to minimize squared loss on two regression datasets, and evaluate both the best-response decision rule and the robust decision rule that results from the self-orthogonality condition of squared error regression. We find that, as predicted by our theory, the robust decision rule outperforms the best-response decision rule under calibration-preserving distribution shift, and that the cost of this robustness is mild even under ideal conditions.

## 1.2 RELATED WORK

Rothblum & Yona (2023) consider a setting in which both the outcome and decision maker's action set are binary, and study how a decision maker should act to minimize their worst case regret over distributions such that the predictor has maximum calibration error bounded by $\alpha$: informally that $|\mathbb{E}[Y|f(x) = v] - v| \leq \alpha$ for all $v$. The models $f$ they study are (approximately) fully calibrated, which is a reasonable assumption in their setting, since they limit their study to 1-dimensional outcomes. In contrast, our interest is not (just) in quantitative measures of full calibration error, but rather qualitatively weaker calibration guarantees, as even approximate full calibration becomes intractable in high dimensions.

A line of recent work (Zhao et al., 2021; Kleinberg et al., 2023; Noarov et al., 2023; Roth & Shi, 2024; Hu & Wu, 2024; Okoroafor et al., 2025) has studied the guarantees that can be given to downstream decision makers who best respond to predictions that have weaker guarantees than full calibration (and which in the cases of Zhao et al. (2021); Noarov et al. (2023); Roth & Shi (2024) can be tractably guaranteed in higher dimensional outcome settings). These guarantees take the form of (external and swap) *regret* bounds, which are qualitatively weaker than the kind of "trustworthiness" promised by full calibration. Informally, regret bounds promise that the decision maker could not have done better by consistently playing a fixed action (or a fixed function remapping their actions to other actions), not that they could not have done better by using a different policy from predictions to actions. We show that even in high dimensions, the tractable "decision calibration" condition given by Zhao et al. (2021) recovers the same "trustworthiness" semantics of full calibration when viewed through our minimax decision making lens.

Analyzing minimax optimal decision policies is a common way of analyzing *robust* or *risk-averse* decision making guarantees, with deep roots in economics (Gilboa & Schmeidler, 1989; Hansen & Sargent, 2001; Manski, 2000; 2004; Manski & Tetenov, 2007; Manski, 2011), statistics (Wald, 1950), and robust optimization (Ben-Tal & Nemirovski, 2002; Kuhn et al., 2019; Duchi & Namkoong, 2021). For example, Carroll (2015) adopts this lens this in the context of contract theory and Kiyani et al. (2025) and Andrews & Chen (2025) do so in the context of conformal prediction. To the best of our knowledge, we are the first to apply this "robust" minimax lens to the problem of partially calibrated high dimensional forecasts.

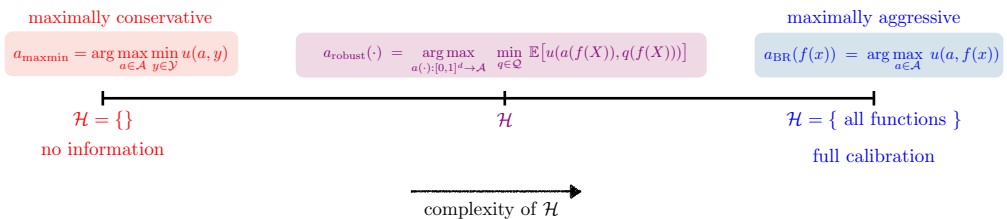

Figure 1: Schematic of the interpolating property

## 2 ROBUST DECISION MAKING AND $\mathcal{H}$-CALIBRATION

In this Section, we define $\mathcal{H}$-calibration as a flexible relaxation of full calibration and then introduce a framework to derive minimax optimal decision making policies that are designed to act on forecasters guaranteed to satisfy $\mathcal{H}$-calibration. This family of calibration guarantees has been studied extensively in the recent literature on multicalibration and its extensions (Hébert-Johnson et al., 2018; Dwork et al., 2021; Gopalan et al., 2022; Deng et al., 2023) — in particular, $\mathcal{H}$-calibration is a special case of what Gopalan et al. (2022) call weighted multicalibration.

**$\mathcal{H}$-Calibration.** Let $\mathcal{H}$ be a set of functions $h : [0,1]^d \to \mathbb{R}$. A forecaster $f$ is said to be $\mathcal{H}$-**calibrated** if for every $h \in \mathcal{H}$,

$$\mathbb{E}\big[\, h(f(X)) \cdot (Y - f(X))\,\big] = 0. \tag{2}$$

Equivalently, writing $q(v) := \mathbb{E}[Y \mid f(X) = v]$ for the true conditional expectation, $\mathcal{H}$-calibration requires

$$\mathbb{E}\big[\, h(f(X)) \cdot (q(f(X)) - f(X))\,\big] = 0, \quad \forall h \in \mathcal{H}. \tag{3}$$

This definition captures a spectrum of guarantees. When $\mathcal{H}$ contains all bounded measurable functions, $\mathcal{H}$-calibration reduces to full calibration — i.e. it requires that $f(v) = q(v) := \mathbb{E}[Y \mid f(X) = v]$ almost surely. For smaller classes $\mathcal{H}$, the requirement is weaker and can be seen as a relaxation of calibration, enforcing consistency only with respect to a restricted set of tests. In the main body of the paper we focus on the $\mathcal{H}$-calibration defined above, but in Appendix B we also discuss scenarios in which only approximate $\mathcal{H}$-calibration is available.

**Robust Decision Making.** Fix an $\mathcal{H}$-calibrated forecaster $f$. Define the set

$$\mathcal{Q} = \Big\{ q : [0,1]^d \to [0,1]^d \ \Big| \ \mathbb{E}\big[\, h(f(X)) \cdot (q(f(X)) - f(X))\,\big] = 0, \ \forall h \in \mathcal{H} \Big\}. \tag{4}$$

In words, $\mathcal{Q}$ consists of all candidate conditional expectations consistent with $f$ satisfying $\mathcal{H}$-calibration. Because the perfect predictor $f(X) = \mathbb{E}[Y|X]$ satisfies $\mathcal{H}$-calibration for every $\mathcal{H}$, the identity map $q(v) = v$ is always in $\mathcal{Q}$—but in general the set may contain many maps. From the perspective of the decision-maker who knows $f$ and the promised calibration guarantee $\mathcal{H}$, but does not know the underlying distribution, given a forecast $f(x)$, the true expectation $\mathbb{E}[Y \mid f(x)]$ is uncertain but must lie within $\mathcal{Q}$. As $\mathcal{H}$ grows richer, $\mathcal{Q}$ shrinks, eventually reducing to $\{q(v) = v\}$ in the case of full calibration.

Faced with this uncertainty, a natural strategy is to adopt a robust policy that guards against the worst-case admissible reality. Formally, the robust decision rule is

$$a_{\text{robust}}(\cdot) = \underset{a(\cdot):[0,1]^d \to \mathcal{A}}{\arg\max} \ \underset{q \in \mathcal{Q}}{\min} \ \mathbb{E}\big[u(a(f(X)), q(f(X)))\big]. \tag{5}$$

That is, the decision-maker chooses an action policy that maximizes utility under the worst-case conditional expectation consistent with calibration guarantees.

**Interpolating Property.** The robust policy in Equation 5 interpolates between two classical extremes (Figure 1). If $\mathcal{H}$ contains all functions, then $\mathcal{Q} = \{q(v) = v\}$ and $a_{\text{robust}}$ reduces to the best response $a_{\text{BR}}(\cdot)$ (Equation equation 1). If $\mathcal{H}$ is empty, then $\mathcal{Q}$ contains all functions and the policy collapses to the constant minimax strategy $a_{\text{Minimax}}(x) = \arg\max_{a \in \mathcal{A}} \min_{y \in [0,1]^d} u(a, y)$. Thus, Equation 5 provides a principled bridge between best-responding to calibrated forecasts and adopting fully conservative policies, with the level of conservatism controlled by the richness of $\mathcal{H}$.

The central theme of the remainder of this paper is to investigate the interaction between different levels of $\mathcal{H}$-calibration and the resulting optimal robust policies. Our focus is not on developing methods for achieving $\mathcal{H}$-calibration itself (for which we refer the reader to a rich line of recent work showing how to accomplish this in both the batch and online adversarial setting Hébert-Johnson et al., 2018; Gopalan et al., 2022; Deng et al., 2023; Noarov et al., 2023; Globus-Harris et al., 2023), but rather on understanding the decision-making consequences once such guarantees are in place. In the next section, we begin by analyzing the general problem of deriving optimal robust decision rules for arbitrary classes $\mathcal{H}$. We then specialize to the important case of decision calibration, showing that this weaker and more practical notion identifies large classes of partially calibrated forecasters for which best responding remains optimal. Beyond its theoretical appeal, this result is also practically useful: when a decision-maker can influence the design or post-processing of the forecaster, they can request a decision-calibrated forecaster, to which they can then simply, reliably, and optimally best respond.

**Assumption 2.1.** The utility $u(a, v)$ is linear in its second argument $v \in [0, 1]^d$ for each $a \in \mathcal{A}$.

This assumption naturally holds in multi-class settings where $v$ is a probability vector over $d$ outcomes and the decision maker has arbitrary utilities $U(a, k)$ for each action–outcome pair. In this case, $u(a, v) = \mathbb{E}[U(a, Y)] = \sum_{k=1}^{d} v_k U(a, k)$, which is linear in $v$. Such risk-neutral expected-utility models underlie much of the calibration and decision-making literature (e.g., (Foster & Vohra, 1997; Kleinberg et al., 2023; Roth & Shi, 2024)). Utilities that are nonlinear in $v$, for example, risk-averse utilities depending on outcome variance, fall outside our framework and represent an important direction for future work.

## 3    OPTIMAL DECISION POLICIES FOR FINITE DIMENSIONAL $\mathcal{H}$-CALIBRATION

In this section, we characterize the optimal robust decision making policies, i.e., solutions to Equation 5. Throughout this section, we assume the function class $\mathcal{H}$ is a finite dimensional space, i.e. it can be described as span of finitely many functions. Formally, let $\mathcal{H} = \mathrm{span}\{h_1, \ldots, h_k\}$ be the linear class generated by measurable $h_i : [0, 1]^d \to \mathbb{R}$. Then the $\mathcal{H}$-calibration condition equation 3 is equivalent to the $k$ linear moment equalities

$$\mathbb{E}\big[h_i(f(X)) \cdot ( q(f(X)) - f(X) )\big] = 0, \qquad i = 1, \ldots, k,$$

so that the ambiguity set in equation 4 may be written as

$$\mathcal{Q} = \Big\{ q : [0, 1]^d \to [0, 1]^d \ \Big| \ \mathbb{E}\big[h_i(f(X)) \cdot ( q(f(X)) - f(X) )\big] = 0 \ \text{ for } i = 1, \ldots, k \Big\}.$$

Intuitively, each equality enforces that, conditional on the forecast, the forecast error has zero correlation with the corresponding test $h_i$; taken together, these constraints exhaust the information provided by $\mathcal{H}$-calibration criteria and hence precisely describe the admissible reality faced by the robust decision-maker in equation 5.

**Theorem 3.1** (Characterization of the Optimal Robust Policy). *Suppose $\mathcal{H} = \mathrm{span}\{h_1, \ldots, h_k\}$ with each $h_i : [0, 1]^d \to \mathbb{R}$, and let $\mathcal{Q}$ be defined as above. Then the minimax problem in Equation 5 admits a saddle point $(a_{\mathrm{robust}}, q^\star)$ with the following structure:*

*There exist multipliers $\lambda^\star = (\lambda_1^\star, \ldots, \lambda_k^\star)$ with each $\lambda_i^\star \in \mathbb{R}^d$ such that for almost every forecast $v = f(x)$ the worst-case map $q^\star(v)$ solves*

$$q^\star(v) \ \in \ \arg\min_{p \in [0,1]^d} \Big\{ \mathrm{val}(p) + p \cdot \sum_{i=1}^{k} h_i(v)\lambda_i^\star \Big\}, \quad \text{where } \mathrm{val}(p) = \max_{a \in \mathcal{A}} u(a, p).$$

*Given $q^\star$, the optimal robust action at $v$ is the best response to $q^\star(v)$:*

$$a_{\mathrm{robust}}(v) \ \in \ \arg\max_{a \in \mathcal{A}} u\big(a, q^\star(v)\big).$$

**Interpretation.** Theorem 3.1 characterizes both the worst-case distribution consistent with $\mathcal{H}$-calibration and the corresponding optimal response. For any realized forecast $\nu = f(x)$, the theorem

yields a simple two-step procedure: compute the adversarial belief

$$q^\star(\nu) \in \arg \min_{p \in [0,1]^d} \{\mathrm{val}(p) + p \cdot s^\star(\nu)\}, \qquad s^\star(\nu) = \sum_{i=1}^k h_i(\nu) \lambda_i^\star,$$

and then take the best response $a_{\mathrm{robust}}(\nu) \in \arg \max_{a \in \mathcal{A}} u(a, q^\star(\nu))$. Thus, the optimal policy is always a best response, not to the raw forecast $f(x)$, but to the adversarially tilted distribution $q^\star(\nu)$ allowed by the calibration constraints. Additionally, a useful consequence is *pointwise computability*: evaluating $a_{\mathrm{robust}}$ at a given $\nu$ reduces to two low-dimensional optimizations, without constructing the full mapping $x \mapsto a_{\mathrm{robust}}(x)$.

From an optimization perspective, the multipliers $\lambda^\star$ solve a finite-dimensional concave maximization problem (see the proof of Theorem 3.1), and $q^\star(\nu)$ is obtained by a pointwise convex minimization over $p \in [0,1]^d$. Both stages can be carried out by standard, fast methods with provable guarantees (e.g., projected subgradient ascent for the dual, or a simple primal–dual scheme), after which one evaluates $q^\star(\nu)$ via the pointwise minimization and takes the best response $a_{\mathrm{robust}}(\nu) = \arg \max_a u(a, q^\star(\nu))$.

In the next section, we analyze the behavior of the resulting decision rules by specializing to concrete $\mathcal{H}$-classes. One might expect that Theorem 3.1 induces a vast hierarchy of policies whose form depends sensitively on $\mathcal{H}$. *Perhaps surprisingly, this is not the case.* In particular, we show a sharp transition: for each decision maker, there exists a specific test class, precisely the one associated with *decision calibration*, such that as soon as $\mathcal{H}$ contains this class, the adversarial tilt collapses ($q^\star(\nu) = \nu$ for a.e. $\nu$) and the optimal robust rule reduces to the plug-in best response to the forecaster.

## 4    ROBUST POLICIES UNDER DECISION CALIBRATION AND BEYOND

In this section, we specialize the general characterization derived in Theorem 3.1 to concrete test classes $\mathcal{H}$. Our core result concerns *decision calibration*: a practically tractable guarantee under which the minimax-optimal robust policy collapses to the plug-in (best-response) rule. This identifies a simple path to decision-theoretic trustworthiness that does not require full calibration.

### 4.1    DECISION CALIBRATION AND PLUG-IN BEST RESPONSE OPTIMALITY

Here we define the variant of decision calibration given by Noarov et al. (2023), a slight strengthening of the definition originally given by Zhao et al. (2021). Fix a single decision problem with action set $\mathcal{A}$ and utility function $u(a, v)$. For each action $a \in \mathcal{A}$, let

$$R_a = \{ v \in [0,1]^d : u(a, v) \geq u(a', v) \text{ for all } a' \in \mathcal{A} \}$$

be the (closed, convex) decision region on which $a$ is a plug-in best response. The *decision-calibration class* is $\mathcal{H}_{\mathrm{dec}} = \{ \mathbf{1}_{R_a} : a \in \mathcal{A} \}$. Here, we denote $\mathbf{1}_A(x) := \mathbf{1}\{x \in A\}$. A forecaster $f$ is *decision calibrated* if it is $\mathcal{H}_{\mathrm{dec}}$-calibrated, i.e.,

$$\mathbb{E}\big[\mathbf{1}_{R_a}\big(f(X)\big)\big(Y - f(X)\big)\big] = 0 \quad \text{for all } a \in \mathcal{A}.$$

Compared to full calibration, decision calibration is far more statistically tractable, since its test class has size $|\mathcal{H}_{\mathrm{dec}}| = |\mathcal{A}|$, a potentially small and fixed number of actions, rather than the large families required for full calibration.

**Theorem 4.1** (Decision calibration $\Rightarrow$ plug-in best response optimality)**.** *If $f$ is $\mathcal{H}_{\mathrm{dec}}$-calibrated, then the minimax-optimal robust rule in equation 5 coincides with the plug-in best response:*

$$a_{\mathrm{robust}}(v) \in \arg \max_{a \in \mathcal{A}} u(a, v) \qquad \text{for almost every } v = f(x).$$

*Equivalently, under decision calibration, best responding to the forecaster is minimax optimal among all forecast-based policies.*

Put differently, upon observing a forecast $v = f(x)$, the decision-maker need only best respond to $v$; no adversarial "tilt" survives the decision-calibration constraints. Conceptually, this upgrades the previously known guarantees of decision calibration—that it implies no swap regret (Noarov et al.,

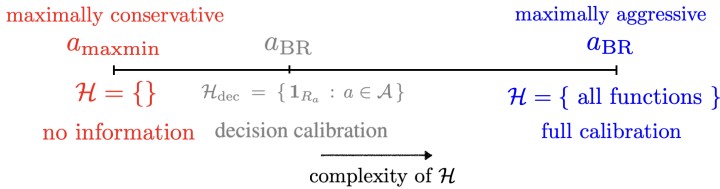

Figure 2: Schematic of the Sharp Transition

2023)—to *minimax optimality*. Swap regret guarantees do not preclude the existence of a policy $a : [0,1]^d \to \mathcal{A}$ that dominates the plugin best response policy $a_{\mathrm{BR}}$ — only that no improved policy has the form $a(v) = \phi(a_{\mathrm{BR}}(v))$ for some mapping $\phi : \mathcal{A} \to \mathcal{A}$, using "actions as a bottleneck". In contrast, Theorem 4.1 directly establishes that no other policy $a : [0,1]^d \to \mathcal{A}$ can improve on the plugin policy $a_{\mathrm{BR}}$ in our minimax sense.

The preceding result assumes that the information conveyed by the forecaster to the decision-maker is exhausted by the decision-calibration tests $\{\mathbf{1}_{R_a}\}_{a \in \mathcal{A}}$. In practice, a forecaster might satisfy additional calibration equalities,

$$\mathbb{E}\big[\, h(f(X)) \cdot \{Y - f(X)\} \,\big] = 0,$$

for functions $h$ beyond the indicators $\mathbf{1}_{R_a}$. The next theorem shows that the plug-in optimality conclusion is stable under such enrichments. This is intuitive: if a forecaster is trustworthy, then making it more calibrated (i.e., adding information) should not diminish that trustworthiness.

**Theorem 4.2.** *Let $\mathcal{H}$ be any test class that contains the decision-calibration indicators, $\mathcal{H}_{\mathrm{dec}} = \{\mathbf{1}_{R_a} : a \in \mathcal{A}\}$. If $f$ is perfectly $\mathcal{H}$-calibrated, then the minimax-optimal robust rule in equation 5 coincides (a.e.) with the plug-in best response:*

$$a_{\mathrm{robust}}(v) \in \arg\max_{a \in \mathcal{A}} u(a,v) \qquad \text{for a.e. } v = f(x).$$

As we make precise in the proof of Theorem 4.2, the "collapse" occurs because the decision-calibration constraints ensure that the expected utility of the plug-in best-response policy $a_{BR}$ is *invariant* to the adversary's choice of $q \in \mathcal{Q}$. For any $q$ satisfying the $\mathcal{H}_{\mathrm{dec}}$ constraints,

$$\mathbb{E}[u(a_{BR}(f(X)), q(f(X)))] = \mathbb{E}[u(a_{BR}(f(X)), f(X))].$$

Thus, the adversary cannot reduce the utility of $a_{BR}$; its worst-case utility equals its nominal utility. Since $a_{BR}$ is the optimal policy under the nominal distribution, and its performance cannot degrade under any admissible $q$, it must also be the minimax-optimal policy.

**Sharp transition.** One might initially expect a *gradual* shift from fully conservative to plug-in best response as $\mathcal{H}$ is enriched. Theorems 4.1–4.2 show a sharper phenomenon (Figure 2): once $\mathcal{H}$ contains the $|\mathcal{A}|$ decision tests $\{\mathbf{1}_{R_a}\}_{a \in \mathcal{A}}$, the adversarial tilt disappears ($q^\star(\nu) = \nu$ a.e.) and the robust rule *collapses* to the plug-in best response equation 1. Enlarging $\mathcal{H}$ further does not change the minimax-optimal policy.

> *Decision calibration is a tractable, task-specific threshold at which robust decision making and plug-in best-response coincide, providing a crisp target for forecaster design and a clear requirement for downstream decision makers.*

As a byproduct, this leads to another practical advantage of decision calibration: a single forecaster can be made simultaneously reliable for a *collection* of downstream decision problems. Intuitively, if the forecast passes the decision calibration tests of each problem, then none of the decision makers needs additional robustness, the plug-in best-response is minimax-optimal for all of them.

**Corollary 4.3** (Simultaneous plug-in optimality across multiple decisions). *Let $u_1, \ldots, u_m$ be $m$ decision problems, with respective action sets $\mathcal{A}_j$ and linear utilities $u_j(a,v)$ in $v \in [0,1]^d$. For each $j$ and $a \in \mathcal{A}_j$, let*

$$R_{a,j} = \{ v \in [0,1]^d : u_j(a,v) \geq u_j(a',v) \, \text{for all } a' \in \mathcal{A}_j \}$$

*be the plug-in decision region of action $a$ in problem $j$, and define the combined test class*

$$\mathcal{H}_{\mathrm{dec}}^{\mathrm{all}} = \bigcup_{j=1}^{m} \{ \mathbf{1}_{R_{a,j}} : a \in \mathcal{A}_j \}.$$

*If $f$ is $\mathcal{H}$-calibrated for some $\mathcal{H}$ satisfying $\mathcal{H}_{\mathrm{dec}}^{\mathrm{all}} \subseteq \mathcal{H}$, then for every $j \in \{1, \ldots, m\}$ the minimax-optimal robust policy for problem $j$ coincides (a.e.) with the plug-in best response:*

$$a_{\mathrm{robust},j}(v) \in \arg\max_{a \in \mathcal{A}_j} u_j(a, v) \qquad \text{for a.e. } v = f(x).$$

*Proof.* For each problem $j$, the included indicators $\{\mathbf{1}_{R_{a,j}}\}_{a \in \mathcal{A}_j}$ ensure that $\mathcal{H}$ contains the decision-calibration tests of problem $j$. Theorem 4.2 then applies verbatim to each $j$, yielding plug-in optimality problem by problem.

## 4.2 BEYOND DECISION CALIBRATION: GENERIC $\mathcal{H}$-CLASSES FROM TRAINING PIPELINES

Thus far we have focused on *decision calibration*, which, when attainable, collapses $a_{\mathrm{robust}}$ to the plug-in best response. In practice, two regimes arise. (i) If one can influence the forecaster's training pipeline, decision calibration is the natural target: it is practical, and our results guarantee plug-in minimax optimality. (ii) If one *cannot* control training, the forecaster might not be decision calibrated for the downstream task. Identifying its partial-calibration profile may be difficult, yet certain moment conditions arise *structurally* from standard training procedures. We give two examples of how to leverage such "free" structure to specify usable $\mathcal{H}$'s and derive the associated robust policies.

**Self-orthogonality from squared-loss training.** A ubiquitous example is *self-orthogonality* (a form of self-calibration) that follows from first-order optimality when a model with a linear last layer is trained to minimize mean squared error. This includes the universally adopted cases of regression with either a linear model or a neural network with a linear head, trained by mean squared error. This and similar guarantees for other loss functions have previously been investigated as consequences of *low degree multicalibration* (Gopalan et al., 2022).

**Proposition 4.4** (Self-orthogonality under squared loss). *Let $X \mapsto z_\phi(X) \in \mathbb{R}^k$ be a representation and $f_\theta(X) = W z_\phi(X) \in \mathbb{R}^d$ a linear last layer. Suppose $\theta = (\phi, W)$ is trained to a first-order stationary point of the expected squared loss*

$$\mathcal{L}(\theta) = \tfrac{1}{2} \mathbb{E}\Big[\big\| f_\theta(X) - Y \big\|_2^2 \Big].$$

*Then the following calibration moments hold:*

$$\mathbb{E}\big[z_\phi(X)\,(Y - f_\theta(X))^\top\big] = 0 \quad \text{and} \quad \mathbb{E}\big[f_\theta(X)\,(Y - f_\theta(X))^\top\big] = 0.$$

*In particular, $f_\theta$ is $\mathcal{H}$-calibrated for the test class $\mathcal{H} = \{h_j(v) = e_j^\top v : j = 1, \ldots, d\}$ (and for any linear combination thereof).*

**Implications.** Proposition 4.4 provides a generic, pipeline-induced $\mathcal{H}$-calibration guarantee whenever a linear head is trained to stationarity under squared loss. Specializing Theorem 3.1 to this setting yields a simple dual. For $d = 1$ (e.g., one-dimensional regression) with $\mathcal{H} = \{h(v) = v\}$, the multiplier is a scalar $\lambda$, and for each forecast $\nu = f(x)$ the worst-case distribution is

$$q^\star(\nu) \in \arg\min_{p \in [0,1]} \{\mathrm{val}(p) + \lambda \nu p\}, \qquad \mathrm{val}(p) = \max_{a \in \mathcal{A}} u(a, p).$$

The robust action is then: $a_{\mathrm{robust}}(\nu) \in \arg\max_{a \in \mathcal{A}} u(a, q^\star(\nu))$. When $u(a, p)$ is linear in $p$ and $\mathcal{A}$ is finite, val is convex piecewise linear, so the inner minimization reduces to checking finitely many candidate points (endpoints and pairwise breakpoints). The dual objective

$$G(\lambda) = \mathbb{E}\Big[\min_{p \in [0,1]} \{\mathrm{val}(p) + \lambda f(X)p\}\Big] - \lambda\, \mathbb{E}[f(X)^2]$$

is concave in $\lambda$ and can be maximized via standard one-dimensional methods (e.g., bisection on a monotone subgradient). In higher dimensions ($d > 1$), the correction term $\lambda \nu p$ becomes $\Lambda \nu p$ for a matrix of multipliers $\Lambda$, and the pointwise problem remains a small convex program over $p \in [0,1]^d$; for finite $\mathcal{A}$ and linear utilities, it is again efficiently solvable.

**Zero-bias and bin-wise calibration.** A widely available source of partial calibration comes from *post-hoc recalibration* that many practitioners already apply (mean correction, histogram binning, isotonic-style step fits on a held-out split). These procedures enforce generic (not task-specific) moment constraints that are directly usable in our framework. We focus on *bin-wise* calibration: take a partition of the forecast range into bins $\{B_1, \ldots, B_J\}$ and enforce, for each bin,

$$\mathbb{E}\Big[\mathbf{1}_{\{f(X) \in B_j\}} (Y - f(X))\Big] = 0, \qquad j = 1, \ldots, J.$$

This corresponds to the test class $\mathcal{H}_{\mathrm{bin}} = \{\mathbf{1}_{B_j} : j = 1, \ldots, J\}$, and reduces to zero-bias when $J=1$ with $B_1 = [0,1]^d$.

**Proposition 4.5** (Robust policy under bin-wise calibration). *Let the utility be linear in the outcome and the action set $\mathcal{A}$ be finite. If $f$ is $\mathcal{H}_{\mathrm{bin}}$-calibrated, then with*

$$m_j := \mathbb{E}[f(X) \mid f(X) \in B_j] = \mathbb{E}[Y \mid f(X) \in B_j],$$

*the worst-case belief is piecewise constant*

$$q^\star(v) = m_j \quad \text{for } v \in B_j \text{ (a.e.)},$$

*and the robust action best-responds to the bin mean:*

$$a_{\mathrm{robust}}(v) \in \arg\max_{a \in \mathcal{A}} \big\{ u(a, m_j) \big\} \qquad \text{for } v \in B_j \text{ (a.e.)}.$$

**Implications.** Bin-wise calibration $\mathcal{H}_{\mathrm{bin}}$ can be obtained cheaply via standard post-hoc methods (histogram binning or isotonic regression), and Proposition 4.5 yields an especially simple, closed-form characterization of the robust policy. Computing $a_{\mathrm{robust}}$ reduces to: (i) estimating $m_j$ on a calibration split, and (ii) at test time, mapping $v$ to its bin $B_j$ and best-responding to $m_j$. No additional optimization is needed to compute actions. As a special case, when $J = 1$ we recover the global-mean constraint $\mathbb{E}[Y - f(X)] = 0$. Then $q^\star$ is constant, $q^\star(v) \equiv \bar{m}$, with $\bar{m} = \mathbb{E}[f(X)] = \mathbb{E}[Y]$, and the robust rule ignores $v$ and plays $\arg\max_{a \in \mathcal{A}} u(a, \bar{m})$. As the partition is refined, the robust rule moves from a single global plug-in best response at $\bar{m}$ to a piecewise plug-in best response at $m_j$, yielding a richer, finer-grained decision policy.

## 5 Experiments

In this section, we evaluate the validity and practical consequences of our framework by implementing our methods on two real-world datasets. We compare the *plug-in best response* ($a_{\mathrm{BR}}$) against the *robust policy* ($a_{\mathrm{robust}}$), which enjoys minimax optimality guarantees under $\mathcal{H}$-calibration.

We focus on two classes of metrics. *Nominal performance* measures average utility when the test data are i.i.d. from the same distribution as the training and calibration splits; this reflects an optimistic regime that often degrades in practice. *Adversarial performance* probes the other extreme by altering the test-time outcome distribution in two ways: (i) a worst case tailored to the plug-in policy, and (ii) a worst case induced by the robust dual, tailored to the robust policy. In both cases, the adversarial distributions respect the $\mathcal{H}$-calibration constraints and are therefore indistinguishable, from the decision-maker's perspective, from i.i.d. test draws given an $\mathcal{H}$-calibrated forecaster.

Our theory predicts two patterns. First, by minimax optimality, the robust policy should dominate the plug-in rule when each is evaluated against its *own* worst-case distribution (and typically also under the adversary tuned to hurt the plug-in). Second, because $(a_{\mathrm{robust}}, q^\star)$ forms a saddle point of equation 5, when both policies are evaluated under the robust-tuned adversary, the robust policy should not underperform the plug-in rule. Under nominal i.i.d. evaluation, the plug-in rule may achieve higher utility, reflecting the lack of need for conservatism in that regime.

### 5.1 Case Studies: Bike Sharing and California Housing

We evaluate our framework on two regression datasets with distinct decision-making interpretations.

**Bike Sharing (UCI).** The UCI *Bike Sharing* (daily) dataset Fanaee-T & Gama (2014) records daily rider counts alongside calendar and weather covariates (season, month, weekday, holiday, working day, weather state, temperature, humidity, wind). The outcome $Y \in [0, 1]$ is the rescaled total rider count, and the decision-maker chooses a staffing/capacity multiplier from $\mathcal{A} = \{0.8, 1.0, 1.2\}$, interpretable as conservative, nominal, and aggressive provisioning.

Table 1: Mean utility on the test set under natural i.i.d. evaluation and two adversarial evaluations. Adversaries respect $\mathcal{H}$-calibration ($\mathcal{H} = \{h(v) = v\}$).

| Dataset | i.i.d. | | Worst-case for robust | | Worst-case for plug-in | |
|---|---|---|---|---|---|---|
| | Plug-in | Robust | Plug-in | Robust | Plug-in | Robust |
| Bike Sharing (UCI) | 0.474 | 0.463 | 0.402 | 0.410 | 0.393 | 0.412 |
| California Housing | 0.216 | 0.207 | 0.160 | 0.164 | 0.155 | 0.166 |

**California Housing.** The *California Housing* dataset Pace & Barry (1997) records median house values (rescaled to $[0, 1]$) with demographic and geographic covariates (median income, housing age, population, latitude/longitude, etc.). Here the decision-maker chooses an investment multiplier from $\mathcal{A} = \{0.6, 0.75, 0.90\}$, interpretable as conservative, nominal, and aggressive investment.

**Utility specification.** In both settings we adopt the utility function $u(a, y) = \alpha\, a\, y - C(a)$, which is linear in $y$. The benefit term $\alpha\, a\, y$ captures service or return proportional to realized outcome $y$, scaled by $\alpha > 0$. The cost term $C(a)$ grows in $a$, penalizing aggressive choices via over-provisioning costs or investment risk. This form tunes the under/over-trade-off without departing from linearity. For Bike Sharing we use $(\alpha, C(\cdot)) = (0.9, \{0.02, 0.05, 0.1\})$, while for California Housing we use $(\alpha, C(\cdot)) = (0.9, \{0.02, 0.05, 0.20\})$. The qualitative conclusions of this Section remain the same under other reasonable parameter choices.

**Forecasting model.** In both datasets, the forecaster $f$ is a two-layer MLP regressor trained to optimize mean squared error. By the self-orthogonality property of linear heads under squared loss (Proposition 4.4), the learned forecaster approximately satisfies $\mathcal{H}$-calibration with $\mathcal{H} = \{h(v) = v\}$, which is the calibration constraint used to derive the robust policy $a_{\text{robust}}$. All experiments use an i.i.d. train/calibration/test split (60/20/20). We use the calibration data to substitute any population level expectation that is needed to be computed to derive $a_{\text{robust}}$.

**Results.** Table 1 reports the mean utilities. The results match theory: under adversaries tailored to the robust policy, the robust rule achieves at least the plug-in performance; under adversaries tuned to harm the plug-in rule, the robust policy secures noticeably higher utility, reflecting its minimax protection. Moreover, the robust policy outperforms the plug-in best response when each is evaluated against its own worst-case distribution.

## 6 Conclusion and Limitations

We developed a decision-theoretic framework for acting on partially calibrated forecasts via a minimax-optimal robust policy over $\mathcal{H}$-calibrated forecasters. We then identified a sharp transition in the behavior of these policies: for any decision problem with $m$ actions, there exist $m$ decision tests (the decision-calibration class) such that, once they are included in $\mathcal{H}$, the robust policy *collapses* to the plug-in best response. This spotlights decision calibration as a natural requirement whenever the decision-maker can influence the training pipeline. Moreover, even when decision calibration is unavailable, we showed that generic properties induced by standard training and post hoc procedures (e.g., self-orthogonality under squared loss and bin-wise calibration) yield usable test classes $\mathcal{H}$ and tractable robust policies within our framework.

Our model assumed that downstream decision makers were risk neutral — i.e., their utility functions $u(a, v)$ are linear in $v$ and $\mathcal{A}$ is finite; these are standard assumptions in the calibration literature, but broadening them would be interesting. We note that certain classes of non-linear utility functions can be linearized over an appropriate basis (Gopalan et al., 2024b; Lu et al., 2025), which would allow our results to apply — though these bases are not always low dimensional enough to be practical.

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

# Appendix

## A  MISSING PROOFS FROM THE MAIN BODY

**Proof of Theorem 3.1**

*Proof.* We begin from the robust formulation

$$\max_{a(\cdot):\mathcal{X}\to\mathcal{A}} \min_{q\in\mathcal{Q}} \mathbb{E}\big[u\big(a(f(X)),\, q(f(X))\big)\big], \tag{6}$$

where $\mathcal{A}\subset\mathbb{R}^m$ is compact, $u(\cdot,\cdot)$ is linear in its second component, $\mathcal{Q}$ is the nonempty, convex, and compact set of measurable maps $q:[0,1]^d\to[0,1]^d$ satisfying the linear moment equalities in equation 4, and $a(\cdot)$ ranges over measurable policies with values in $\mathcal{A}$. The mapping $(a,q)\mapsto \mathbb{E}[u(a(f(X)),q(f(X)))]$ is convex in $q$ (since $u(a,\cdot)$ is linear, hence convex, in $y$ and expectation preserves convexity), concave in $a$ (as a pointwise maximum over linear functionals in $a$ on the compact set $\mathcal{A}$). Hence, by Sion's minimax theorem,

$$\max_{a(\cdot)} \min_{q\in\mathcal{Q}} \mathbb{E}\big[u(a(f(X)),q(f(X)))\big] \;=\; \min_{q\in\mathcal{Q}} \max_{a(\cdot)} \mathbb{E}\big[u(a(f(X)),q(f(X)))\big].$$

Fix any $q\in\mathcal{Q}$. The inner maximization over policies separates pointwise in $v=f(x)$, yielding the value function

$$\mathrm{val}(p) \;\triangleq\; \max_{a\in\mathcal{A}} u(a,p) \quad\text{and}\quad \max_{a(\cdot)} \mathbb{E}\big[u\big(a(f(X)),q(f(X))\big)\big] = \mathbb{E}\big[\mathrm{val}\big(q(f(X))\big)\big].$$

Therefore the robust value equals the convex adversarial problem

$$\min_{q\in\mathcal{Q}} \mathbb{E}\big[\mathrm{val}\big(q(f(X))\big)\big], \tag{7}$$

which will be analyzed via Lagrangian duality below.

Introduce vector Lagrange multipliers $\lambda_i\in\mathbb{R}^d$ for the $d$-dimensional equalities in equation 4, and let $\lambda=(\lambda_1,\dots,\lambda_k)$. Define

$$s(v) \;\triangleq\; \sum_{i=1}^{k} h_i(v)\,\lambda_i \;\in\; \mathbb{R}^d, \qquad v\in[0,1]^d.$$

The Lagrangian of equation 7 is

$$L(q,\lambda) \;=\; \mathbb{E}\big[\mathrm{val}\big(q(f(X))\big)\big] \;+\; \sum_{i=1}^{k}\lambda_i\cdot\mathbb{E}\big[h_i\big(f(X)\big)\big(q(f(X))-f(X)\big)\big].$$

By linearity of expectation,

$$L(q,\lambda) \;=\; \mathbb{E}\Big[\mathrm{val}\big(q(f(X))\big) \;+\; q(f(X))\cdot s\big(f(X)\big) \;-\; f(X)\cdot s\big(f(X)\big)\Big].$$

The dual function is obtained by minimizing $L(q,\lambda)$ over measurable $q:[0,1]^d\to[0,1]^d$. Since the integrand depends on $q$ only through $q(f(X))$, the infimum can be taken *pointwise* in the forecast value $v=f(X)$:

$$G(\lambda) \;=\; \inf_q L(q,\lambda) \;=\; \mathbb{E}\Big[\inf_{p\in[0,1]^d}\big\{\mathrm{val}(p)+p\cdot s\big(f(X)\big)\big\}\Big] \;-\; \mathbb{E}\big[f(X)\cdot s\big(f(X)\big)\big].$$

The primal problem equation 7 is convex (convex objective, affine constraints) and feasible (e.g., $q(v)=v$), thereby strong duality holds. Hence,

$$\min_{q\in\mathcal{Q}}\mathbb{E}\big[\mathrm{val}\big(q(f(X))\big)\big] \;=\; \max_{\lambda\in(\mathbb{R}^d)^k} G(\lambda),$$

and there exists a maximizing multiplier $\lambda^\star$. Define

$$s^\star(v) \;\triangleq\; \sum_{i=1}^{k} h_i(v)\,\lambda_i^\star \in \mathbb{R}^d.$$

By the definition of $G(\lambda)$ and strong duality, any primal optimizer $q^\star \in \mathcal{Q}$ must minimize the Lagrangian at $\lambda^\star$. Since the dependence on $q$ is only through $q(f(X))$, this yields the pointwise characterization, for $v = f(x)$ almost surely,

$$q^\star(v) \in \arg\min_{p \in [0,1]^d} \left\{ \mathrm{val}(p) + p \cdot s^\star(v) \right\}.$$

With $q^\star$ fixed, define the policy

$$a_{\mathrm{robust}}(v) \in \arg\max_{a \in \mathcal{A}} u\big(a, q^\star(v)\big).$$

Then, by the definition of val and the construction of $q^\star$,

$$\max_{a(\cdot)} \mathbb{E}\big[u\big(a(f(X)), q^\star(f(X))\big)\big] = \mathbb{E}\big[\mathrm{val}\big(q^\star(f(X))\big)\big] = \min_{q \in \mathcal{Q}} \mathbb{E}\big[\mathrm{val}\big(q(f(X))\big)\big],$$

which shows that $(a_{\mathrm{robust}}, q^\star)$ is a saddle point of equation 6. In particular, $a_{\mathrm{robust}}$ is optimal for the outer maximization, and $q^\star$ is worst–case optimal for the inner minimization, with $q^\star$ characterized pointwise by the minimization problem above and determined by the dual multiplier $\lambda^\star$. This matches the statement of Theorem 3.1 and completes the proof. □

**Proof of Theorem 4.1**:

*Proof.* We use the reduction

$$\max_{a(\cdot)} \min_{q \in \mathcal{Q}} \mathbb{E}\big[u(a(f(X)), q(f(X)))\big] = \min_{q \in \mathcal{Q}} \mathbb{E}\big[\mathrm{val}(q(f(X)))\big],$$

established in the proof of Theorem 3.1. Fix the decision regions

$$R_a = \{ v \in [0,1]^d : u(a,v) \geq u(a',v) \, \forall a' \in \mathcal{A} \},$$

each convex. Under $\mathcal{H}_{\mathrm{dec}} = \{\mathbf{1}_{R_a} : a \in \mathcal{A}\}$, admissible $q$ satisfy

$$\mathbb{E}\big[\mathbf{1}_{R_a}(f(X))\{q(f(X)) - f(X)\}\big] = 0 \quad \forall a,$$

equivalently (whenever $\mathbb{P}(f(X) \in R_a) > 0$),

$$\mathbb{E}[q(f(X)) \mid f(X) \in R_a] = \mathbb{E}[f(X) \mid f(X) \in R_a] =: \mu_a \in R_a.$$

By Jensen's inequality (convexity of val), for any $q \in \mathcal{Q}$ and any $a$,

$$\mathbb{E}\big[\mathrm{val}\big(q(f(X))\big) \mid f(X) \in R_a\big] \geq \mathrm{val}(\mu_a).$$

Define the piecewise-constant $\bar{q}(v) = \sum_a \mu_a \, \mathbf{1}_{R_a}(v)$. Then $\bar{q} \in \mathcal{Q}$ and, conditionally on $f(X) \in R_a$, we have $\bar{q}(f(X)) = \mu_a$ a.s., hence the bound is attained:

$$\mathbb{E}\big[\mathrm{val}\big(\bar{q}(f(X))\big)\big] = \sum_a \mathbb{P}(f(X) \in R_a)\,\mathrm{val}(\mu_a) \leq \mathbb{E}\big[\mathrm{val}\big(q(f(X))\big)\big] \quad \forall q \in \mathcal{Q}.$$

Thus a worst-case belief is $q^\star = \bar{q}$, region-wise constant with $q^\star(v) = \mu_a$ on $R_a$.

Finally, since $\mu_a \in R_a$, by definition of $R_a$ we have $u(a, \mu_a) \geq u(a', \mu_a)$ for all $a'$, so $a$ is a best response to $\mu_a$. Therefore the robust action at $v \in R_a$ is

$$a_{\mathrm{robust}}(v) \in \arg\max_{a'} u(a', q^\star(v)) = \arg\max_{a'} u(a', \mu_a) \ni a,$$

which coincides (a.e.) with the plug-in best response to $v$. This proves Theorem 4.1. □

**Proof of Theorem 4.2:**

Recall $\mathrm{val}(p) = \max_{a \in \mathcal{A}} u(a, p)$ and the reduction

$$\max_{a(\cdot)} \min_{q \in \mathcal{Q}_\mathcal{H}} \mathbb{E}\big[u\big(a(f(X)), q(f(X))\big)\big] = \min_{q \in \mathcal{Q}_\mathcal{H}} \mathbb{E}\big[\mathrm{val}\big(q(f(X))\big)\big],$$

established earlier in the proof of Theorem 3.1. Moreover, the identity map $q_{\mathrm{id}}(v) = v$ always lies in $\mathcal{Q}_{\mathcal{H}}$ (the perfect forecaster is consistent with every $\mathcal{H}$-calibration constraint), so for any policy $a(\cdot)$,

$$\min_{q \in \mathcal{Q}_{\mathcal{H}}} \mathbb{E}\big[u\big(a(f(X)), q(f(X))\big)\big] \ \leq \ \mathbb{E}\big[u\big(a(f(X)), f(X)\big)\big]. \tag{8}$$

Let $a_{\mathrm{BR}}(v) \in \arg\max_{a \in \mathcal{A}} u(a, v)$ be a plug-in best response.[1] We show that, assuming $\mathcal{H}$ contains the decision-calibration tests $\{\mathbf{1}_{R_a}\}_{a \in \mathcal{A}}$,

$$\mathbb{E}\big[u\big(a_{\mathrm{BR}}(f(X)), q(f(X))\big)\big] \ = \ \mathbb{E}\big[u\big(a_{\mathrm{BR}}(f(X)), f(X)\big)\big] \qquad \forall q \in \mathcal{Q}_{\mathcal{H}}. \tag{9}$$

Write $\mu_a := \mathbb{E}[\, f(X) \mid f(X) \in R_a \,]$ whenever $\mathbb{P}(f(X) \in R_a) > 0$ (if $\mathbb{P}(f(X) \in R_a) = 0$, any choice of $\mu_a$ is harmless since the corresponding terms vanish). Then

$$
\begin{aligned}
\mathbb{E}\big[u\big(a_{\mathrm{BR}}(f(X)), q(f(X))\big)\big] &= \sum_{a \in \mathcal{A}} \mathbb{E}\big[u\big(a, q(f(X))\big) \, \mathbf{1}_{\{f(X) \in R_a\}}\big] \\
&\overset{(i)}{=} \sum_{a \in \mathcal{A}} \mathbb{P}(f(X) \in R_a) \, u\big(a, \, \mathbb{E}[q(f(X)) \mid f(X) \in R_a]\big) \\
&\overset{(ii)}{=} \sum_{a \in \mathcal{A}} \mathbb{P}(f(X) \in R_a) \, u\big(a, \, \mathbb{E}[f(X) \mid f(X) \in R_a]\big) \\
&= \sum_{a \in \mathcal{A}} \mathbb{P}(f(X) \in R_a) \, u(a, \mu_a) \\
&\overset{(iii)}{=} \sum_{a \in \mathcal{A}} \mathbb{E}\big[u\big(a, f(X)\big) \, \mathbf{1}_{\{f(X) \in R_a\}}\big] \\
&= \mathbb{E}\big[u\big(a_{\mathrm{BR}}(f(X)), f(X)\big)\big].
\end{aligned}
$$

Here: $(i)$ uses that $u(a, \cdot)$ is linear in its second argument, so

$$\mathbb{E}[u(a, q(f(X))) \mid f(X) \in R_a] = u(a, \mathbb{E}[q(f(X)) \mid f(X) \in R_a]),$$

$(ii)$ uses the decision-calibration equalities $\mathbb{E}[\mathbf{1}_{R_a}(f(X))\{q(f(X)) - f(X)\}] = 0$, equivalently $\mathbb{E}[q(f(X)) \mid f(X) \in R_a] = \mathbb{E}[f(X) \mid f(X) \in R_a] = \mu_a$ whenever $\mathbb{P}(f(X) \in R_a) > 0$; and $(iii)$ again uses linearity: $u(a, \mu_a) = u\big(a, \mathbb{E}[f(X) \mid f(X) \in R_a]\big) = \mathbb{E}[u(a, f(X)) \mid f(X) \in R_a]$.

Combining equation 8, the optimality of best response on the *perceived* outcomes,

$$\mathbb{E}\big[u\big(a(f(X)), f(X)\big)\big] \ \leq \ \mathbb{E}\big[u\big(a_{\mathrm{BR}}(f(X)), f(X)\big)\big] \qquad \text{for all policies } a(\cdot),$$

and the invariance equation 9, we obtain the minimax dominance

$$\min_{q \in \mathcal{Q}_{\mathcal{H}}} \mathbb{E}\big[u\big(a_{\mathrm{BR}}(f(X)), q(f(X))\big)\big] \ = \ \mathbb{E}\big[u\big(a_{\mathrm{BR}}(f(X)), f(X)\big)\big] \ \geq \ \min_{q \in \mathcal{Q}_{\mathcal{H}}} \mathbb{E}\big[u\big(a(f(X)), q(f(X))\big)\big],$$

for every forecast-based policy $a(\cdot)$. Hence the plug-in best response is minimax optimal under any $\mathcal{H}$ that contains the decision-calibration tests, as claimed.

**Proof of Proposition 4.4**:

*Proof.* Assume $\mathbb{E}\|z_\phi(X)\|_2^2 < \infty$ and $\mathbb{E}\|Y\|_2^2 < \infty$ so that all derivatives and expectations below are well-defined and we may interchange expectation and differentiation by dominated convergence. Write $z := z_\phi(X) \in \mathbb{R}^k$ and $f := f_\theta(X) = Wz \in \mathbb{R}^d$. The squared-loss risk is

$$\mathcal{L}(\theta) \ = \ \tfrac{1}{2} \, \mathbb{E}\big[\|f - Y\|_2^2\big] \ = \ \tfrac{1}{2} \, \mathbb{E}\big[(Wz - Y)^\top (Wz - Y)\big].$$

For the linear head $W \in \mathbb{R}^{d \times k}$, the gradient with respect to $W$ satisfies the standard identity

$$\nabla_W \left( \tfrac{1}{2} \|Wz - Y\|_2^2 \right) \ = \ (Wz - Y) \, z^\top \in \mathbb{R}^{d \times k}.$$

Taking expectation and interchanging $\nabla$ with $\mathbb{E}$ yields

$$\nabla_W \mathcal{L}(\theta) \ = \ \mathbb{E}\big[(f - Y) \, z^\top\big].$$

---

[1]Fix any deterministic tie-breaking so that $a_{\mathrm{BR}}$ and the regions $R_a = \{v : a_{\mathrm{BR}}(v) = a\}$ are measurable.

At a first-order stationary point (in particular, when the gradient with respect to $W$ vanishes) we have
$$\mathbb{E}\big[(f - Y)\,z^\top\big] \;=\; 0_{d \times k}.$$

Transposing gives
$$\mathbb{E}\big[z\,(f - Y)^\top\big] \;=\; 0_{k \times d} \qquad \Longleftrightarrow \qquad \mathbb{E}\big[z\,(Y - f)^\top\big] \;=\; 0_{k \times d},$$
which is the first claimed moment identity.

For the second identity, observe that $f = Wz$, hence
$$\mathbb{E}\big[f\,(Y - f)^\top\big] \;=\; \mathbb{E}\big[Wz\,(Y - f)^\top\big] \;=\; W\,\mathbb{E}\big[z\,(Y - f)^\top\big] \;=\; W\,0_{k \times d} \;=\; 0_{d \times d}.$$

Therefore both $\mathbb{E}[\,z_\phi(X)\,(Y - f_\theta(X))^\top] = 0$ and $\mathbb{E}[\,f_\theta(X)\,(Y - f_\theta(X))^\top] = 0$ hold. In particular, for each coordinate $j = 1, \ldots, d$, $\mathbb{E}[\,e_j^\top f_\theta(X)\,(Y - f_\theta(X))^\top] = 0$ and $\mathbb{E}[\,z_\phi(X)\,e_j^\top(Y - f_\theta(X))] = 0$, so $f_\theta$ is $\mathcal{H}$-calibrated for $\mathcal{H} = \{h_j(v) = e_j^\top v : \; j = 1, \ldots, d\}$ and for any linear combination thereof. This proves the proposition. $\qquad\square$

**Proof of Proposition 4.5:**

*Proof.* By the reduction established earlier (see the proof of Theorem 3.1), the robust problem
$$\max_{a(\cdot)} \min_{q \in \mathcal{Q}} \mathbb{E}\big[u(a(f(X)), q(f(X)))\big]$$
with linear utilities and finite $\mathcal{A}$ is equivalent to the convex program
$$\min_{q \in \mathcal{Q}} \mathbb{E}\big[\mathrm{val}\big(q(f(X))\big)\big], \qquad \mathrm{val}(p) := \max_{a \in \mathcal{A}} u(a, p),$$
subject to the $\mathcal{H}_{\mathrm{bin}}$-calibration constraints
$$\mathbb{E}\big[\mathbf{1}_{\{f(X) \in B_j\}}\,(q(f(X)) - f(X))\big] = 0, \qquad j = 1, \ldots, J.$$
Write $E_j := \{f(X) \in B_j\}$ and assume $\mathbb{P}(E_j) > 0$ (bins with zero probability are immaterial). Then the constraints are equivalent to
$$\mathbb{E}[q(f(X)) \,|\, E_j] \;=\; \mathbb{E}[f(X) \,|\, E_j] \;=:\; m_j, \qquad j = 1, \ldots, J.$$
Because $u(a, \cdot)$ is linear in the outcome, $\mathrm{val}$ is the pointwise maximum of linear maps and hence convex. Decomposing by bins and applying Jensen's inequality gives, for any feasible $q$,
$$\mathbb{E}\big[\mathrm{val}\big(q(f(X))\big)\big] = \sum_{j=1}^{J} \mathbb{P}(E_j)\,\mathbb{E}\big[\mathrm{val}\big(q(f(X))\big) \,\big|\, E_j\big]$$
$$\geq \sum_{j=1}^{J} \mathbb{P}(E_j)\,\mathrm{val}\big(\mathbb{E}[q(f(X)) \,|\, E_j]\big)$$
$$= \sum_{j=1}^{J} \mathbb{P}(E_j)\,\mathrm{val}(m_j).$$

Define the piecewise-constant candidate
$$\bar{q}(v) \;:=\; \sum_{j=1}^{J} m_j\,\mathbf{1}_{B_j}(v).$$
Then $\bar{q}$ is feasible, since for each $j$,
$$\mathbb{E}\big[\mathbf{1}_{E_j}\,(\bar{q}(f(X)) - f(X))\big] = \mathbb{P}(E_j)\,\big(m_j - \mathbb{E}[f(X) \,|\, E_j]\big) = 0,$$
and it attains the Jensen lower bound because $\bar{q}(f(X)) = m_j$ almost surely on $E_j$:
$$\mathbb{E}\big[\mathrm{val}\big(\bar{q}(f(X))\big) \,\big|\, E_j\big] = \mathrm{val}(m_j).$$
Therefore $\bar{q}$ is an optimizer, and any minimizer $q^\star$ can be chosen (a.e.) piecewise constant with $q^\star(v) = m_j$ for $v \in B_j$.

Finally, fixing such a $q^\star$, the robust action at forecast $v \in B_j$ solves
$$a_{\mathrm{robust}}(v) \in \arg\max_{a \in \mathcal{A}} u\big(a, q^\star(v)\big) = \arg\max_{a \in \mathcal{A}} u\big(a, m_j\big),$$
which depends only on the bin index, i.e., it is the best response to the bin mean. This proves the claim. $\qquad\square$

# B    APPROXIMATE $\mathcal{H}$-CALIBRATION: STABILITY UNDER $\varepsilon$-SLACK

This appendix extends the main results to the practically relevant regime in which $\mathcal{H}$-calibration holds only approximately. Concretely, we relax each linear calibration equality in equation 3 to an $\ell_2$–ball of radius $\varepsilon$. Throughout, we retain the standing assumptions of the main text: utilities are linear in the outcome, so there exist $\{r_a \in \mathbb{R}^d,\ c_a \in \mathbb{R}\}_{a \in \mathcal{A}}$ with

$$u(a, p) \;=\; r_a \cdot p \;+\; c_a \quad \Longrightarrow \quad \mathrm{val}(p) \;:=\; \max_{a \in \mathcal{A}} u(a, p) \text{ is convex and } L\text{-Lipschitz w.r.t. } \|\cdot\|_2,$$

where $L := \max_{a \in \mathcal{A}} \|r_a\|_2$. We write expectations over $(X, Y)$ distributed as in the main body, and $f : \mathcal{X} \to [0, 1]^d$ denotes the given forecaster.

**Approximate calibration constraints.** Let $\mathcal{H} = \mathrm{span}\{h_1, \ldots, h_k\}$ with measurable $h_i : [0, 1]^d \to \mathbb{R}$ bounded by $|h_i(v)| \le 1$. For a candidate conditional expectation $q : [0, 1]^d \to [0, 1]^d$, define the (vector) calibration moments

$$m_i(q) \;:=\; \mathbb{E}\big[\, h_i(f(X)) \big\{ q(f(X)) - f(X) \big\} \,\big] \in \mathbb{R}^d, \qquad i = 1, \ldots, k.$$

We say $q$ is $\varepsilon$–*approximately $\mathcal{H}$-calibrated* if $\|m_i(q)\|_2 \le \varepsilon$ for all $i$. The corresponding ambiguity set and robust value are

$$\mathcal{Q}_\varepsilon \;:=\; \Big\{ q : [0, 1]^d \to [0, 1]^d \;:\; \|m_i(q)\|_2 \le \varepsilon,\ i = 1, \ldots, k \Big\}, \qquad V_\varepsilon \;:=\; \min_{q \in \mathcal{Q}_\varepsilon} \mathbb{E}\big[\mathrm{val}\big(q(f(X))\big)\big].$$

For reference, the exact-calibration value is $V_0 = \min_{q \in \mathcal{Q}} \mathbb{E}[\mathrm{val}(q(f(X)))]$, where $\mathcal{Q}$ is the equality-based set from equation 4.

**Roadmap.** We first show a *dual penalty* bound: moving from exact to $\varepsilon$–approximate constraints subtracts an explicit $\ell_2$–norm penalty from the exact dual objective, yielding two-sided value bounds and a linear-in-$\varepsilon$ degradation guarantee. We then quantify the robustness of *decision calibration*: even under $\varepsilon$–slack, the plug-in best response is $O(mL\varepsilon)$–minimax optimal (with $m := |\mathcal{A}|$). Finally, for *bin-wise* (histogram) calibration with $\varepsilon$–slack, we obtain piecewise-constant worst-case beliefs and tight value bounds, recovering the exact structural picture up to $O(JL\varepsilon)$ terms when there are $J$ bins.

**Policy characterization under $\varepsilon$–slack.** The primal inner problem remains convex and pointwise in $v = f(x)$, while the dual acquires the norm penalty from Theorem B.1. Consequently, the optimal robust policy admits the same form as in the exact case, with the unique change that the dual multiplier solves a penalized maximization.

**Theorem B.1** ($\varepsilon$–robust policy via penalized dual)*. Let $\mathcal{H} = \mathrm{span}\{h_1, \ldots, h_k\}$ and define $G(\lambda)$ as in the main text. Let*

$$\lambda_\varepsilon^\star \;\in\; \arg \max_{\lambda \in (\mathbb{R}^d)^k} \Big\{ G(\lambda) \;-\; \varepsilon \sum_{i=1}^k \|\lambda_i\|_2 \Big\}, \qquad s_{\lambda_\varepsilon^\star}(v) \;:=\; \sum_{i=1}^k h_i(v)\, \lambda_{\varepsilon, i}^\star.$$

*Then there exists a worst-case belief $q_\varepsilon^\star : [0, 1]^d \to [0, 1]^d$ such that for almost every forecast $v = f(x)$,*

$$q_\varepsilon^\star(v) \;\in\; \arg \min_{p \in [0, 1]^d} \Big\{ \mathrm{val}(p) \;+\; p \cdot s_{\lambda_\varepsilon^\star}(v) \Big\}, \qquad \mathrm{val}(p) = \max_{a \in \mathcal{A}} u(a, p).$$

*The $\varepsilon$–robust action is the best response to $q_\varepsilon^\star(v)$:*

$$a_\varepsilon^\star(v) \;\in\; \arg \max_{a \in \mathcal{A}} u\big(a, q_\varepsilon^\star(v)\big).$$

*Proof of Theorem B.1.* Recall the robust formulation under linear utilities and forecast–based policies reduces to the adversarial convex program

$$\min_{q \in \mathcal{Q}_\varepsilon} \mathbb{E}\big[\mathrm{val}\big(q(f(X))\big)\big], \qquad \mathrm{val}(p) := \max_{a \in \mathcal{A}} u(a, p),$$

with the $\varepsilon$–approximate $\mathcal{H}$–calibration set

$$\mathcal{Q}_\varepsilon = \Big\{ q : [0, 1]^d \to [0, 1]^d :\ \big\| \mathbb{E}[h_i(f(X))\{q(f(X)) - f(X)\}] \big\|_2 \le \varepsilon,\ i = 1, \ldots, k \Big\}.$$

Introduce slack vectors $s_i \in \mathbb{R}^d$ (one per test) so that each constraint is rewritten as the *equality*

$$\mathbb{E}[h_i(f(X))\{q(f(X)) - f(X)\}] = s_i \quad \text{with} \quad \|s_i\|_2 \le \varepsilon \quad (i = 1, \ldots, k).$$

Let $\lambda_i \in \mathbb{R}^d$ be the Lagrange multipliers for these equalities and set $s_\lambda(v) := \sum_{i=1}^k h_i(v)\lambda_i$. The Lagrangian reads

$$L(q, s; \lambda) = \mathbb{E}\Big[\mathrm{val}\big(q(f(X))\big)\Big] + \sum_{i=1}^k \lambda_i \cdot \Big(\mathbb{E}[h_i(f(X))\{q(f(X)) - f(X)\}] - s_i\Big).$$

Minimizing $L$ over the slacks $s_i$ subject to $\|s_i\|_2 \le \varepsilon$ contributes the support function of the $\ell_2$–ball,

$$\inf_{\|s_i\|_2 \le \varepsilon} (-\lambda_i \cdot s_i) = -\sup_{\|s_i\|_2 \le \varepsilon} (\lambda_i \cdot s_i) = -\varepsilon \|\lambda_i\|_2.$$

Minimizing the remaining part over $q$ depends on $q$ only through $q(f(X))$ and yields, pointwise in $v = f(X)$,

$$\inf_q L(q, s; \lambda) = \mathbb{E}\Big[\inf_{p \in [0,1]^d} \{\mathrm{val}(p) + p \cdot s_\lambda(f(X))\}\Big] - \mathbb{E}[f(X) \cdot s_\lambda(f(X))] - \varepsilon \sum_{i=1}^k \|\lambda_i\|_2.$$

Therefore the dual function is

$$G_\varepsilon(\lambda) = \underbrace{\mathbb{E}\Big[\min_{p \in [0,1]^d} \{\mathrm{val}(p) + p \cdot s_\lambda(f(X))\}\Big] - \mathbb{E}[f(X) \cdot s_\lambda(f(X))]}_{=:G(\lambda)} - \varepsilon \sum_{i=1}^k \|\lambda_i\|_2,$$

i.e., the exact-calibration dual $G(\lambda)$ penalized by $\varepsilon \sum_i \|\lambda_i\|_2$.

The primal problem is convex (convex objective, affine moment constraints) and feasible (e.g., $q(v) \equiv v$ makes all moments 0, which is strictly feasible when $\varepsilon > 0$), so Slater's condition holds; hence strong duality holds and a maximizer $\lambda^\star$ of $G_\varepsilon$ exists:

$$\min_{q \in \mathcal{Q}_\varepsilon} \mathbb{E}\big[\mathrm{val}\big(q(f(X))\big)\big] = \max_{\lambda \in (\mathbb{R}^d)^k} \Big\{G(\lambda) - \varepsilon \sum_{i=1}^k \|\lambda_i\|_2\Big\}.$$

Moreover, comparing with the exact case (which corresponds to $\varepsilon = 0$) gives the two-sided value bound

$$\max_\lambda \Big\{G(\lambda) - \varepsilon \sum_i \|\lambda_i\|_2\Big\} \le V_\varepsilon \le V_0 := \max_\lambda G(\lambda),$$

and $0 \le V_0 - V_\varepsilon \le \varepsilon \min_{\lambda \in \arg\max G} \sum_i \|\lambda_i\|_2$.

By strong duality, any primal minimizer $q_\varepsilon^\star \in \mathcal{Q}_\varepsilon$ together with $\lambda^\star$ forms a saddle point: $L(q_\varepsilon^\star, \lambda) \le L(q_\varepsilon^\star, \lambda^\star) \le L(q, \lambda^\star)$. The first inequality implies that $q_\varepsilon^\star$ minimizes the Lagrangian at $\lambda^\star$, which (by the pointwise structure above) yields, for almost every forecast $v = f(x)$,

$$q_\varepsilon^\star(v) \in \arg\min_{p \in [0,1]^d} \Big\{\mathrm{val}(p) + p \cdot s_{\lambda^\star}(v)\Big\}, \qquad s_{\lambda^\star}(v) = \sum_{i=1}^k h_i(v)\lambda_i^\star.$$

With $q_\varepsilon^\star$ fixed, the optimal robust action at $v$ solves

$$a_{\mathrm{robust}, \varepsilon}(v) \in \arg\max_{a \in \mathcal{A}} u\big(a, q_\varepsilon^\star(v)\big),$$

i.e., it is the best response to the worst-case belief $q_\varepsilon^\star(v)$. This is the same best-response structure as in the exact case, now using the penalized dual optimizer $\lambda^\star$ (cf. the exact characterization in the main text).

Altogether, we have (i) the dual penalty representation with value bounds, (ii) existence of a dual maximizer $\lambda^\star$, (iii) the pointwise form of the worst-case belief $q_\varepsilon^\star$, and (iv) the robust policy as a pointwise best response to $q_\varepsilon^\star$, completing the proof. □

**Computation.** Algorithmically, the recipe mirrors the exact case: (i) maximize the concave objective $G(\lambda) - \varepsilon \sum_i \|\lambda_i\|_2$ (e.g., projected/subgradient or bisection in 1D; small-scale mirror descent otherwise); (ii) for each forecast $v$, compute $q_\varepsilon^\star(v)$ by solving the convex problem in $p$; (iii) play $a_\varepsilon^\star(v)$ as the best response to $q_\varepsilon^\star(v)$. For finite $\mathcal{A}$ and utilities linear in $p$, step (ii) reduces to checking a small finite set of candidates (endpoints and pairwise breakpoints of val), exactly as in the main text.

**Decision tests contained in $\mathcal{H}$ under $\varepsilon$–slack: near-optimality of plug-in.** Let $R_a := \{v : u(a, v) \geq u(a', v) \ \forall a' \in \mathcal{A}\}$ be the plug-in region for action $a$, and write $P_a := \mathbb{P}(f(X) \in R_a)$ (regions with $P_a = 0$ are ignorable). Assume $\mathcal{H}$ is a test class that *contains the decision indicators* $\{\mathbf{1}_{R_a} : a \in \mathcal{A}\}$, with each test bounded by $\|\mathbf{1}_{R_a}\|_\infty \leq 1$. We impose $\varepsilon$–approximate $\mathcal{H}$–calibration in the componentwise sense of Section B, so in particular

$$\left\| \mathbb{E}\big[\mathbf{1}_{R_a}(f(X)) \{q(f(X)) - f(X)\}\big] \right\|_2 \leq \varepsilon, \qquad \text{for all } a \in \mathcal{A} \text{ and all } q \in \mathcal{Q}_\varepsilon.$$

**Theorem B.2** (Plug-in is $O(mL\varepsilon)$–minimax optimal when decision tests lie in $\mathcal{H}$). *Let $m := |\mathcal{A}|$ and $L := \max_{a \in \mathcal{A}} \|r_a\|_2$ as above. If $\mathcal{H}$ contains the decision indicators $\{\mathbf{1}_{R_a}\}$ and $f$ is $\varepsilon$–approximately $\mathcal{H}$–calibrated, then the plug-in rule $a_{\mathrm{BR}}(v) \in \arg\max_a u(a, v)$ satisfies, for any forecast-based policy $a(\cdot)$,*

$$\min_{q \in \mathcal{Q}_\varepsilon} \mathbb{E}\big[u(a_{\mathrm{BR}}(f(X)), q(f(X)))\big] \geq \min_{q \in \mathcal{Q}_\varepsilon} \mathbb{E}\big[u(a(f(X)), q(f(X)))\big] - mL\varepsilon.$$

*Proof.* Fix any $q \in \mathcal{Q}_\varepsilon$. Decompose by plug-in regions:

$$\mathbb{E}\big[u(a_{\mathrm{BR}}(f), q(f))\big] = \sum_{a \in \mathcal{A}} P_a \, \mathbb{E}\big[u(a, q(f)) \mid f \in R_a\big].$$

Since $u(a, \cdot)$ is linear,

$$\mathbb{E}[u(a, q(f)) \mid f \in R_a] = u\Big(a, \, \mathbb{E}[q(f) \mid f \in R_a]\Big).$$

Let $\mu_a := \mathbb{E}[f \mid f \in R_a]$. By $L$–Lipschitzness of $u(a, \cdot)$ and the $\varepsilon$–slack on the indicator test,

$$\left| u\big(a, \mathbb{E}[q(f) \mid R_a]\big) - u\big(a, \mu_a\big) \right| \leq L \left\| \mathbb{E}[q(f) - f \mid R_a] \right\|_2 = L \frac{\left\| \mathbb{E}[\mathbf{1}_{R_a}(q(f) - f)] \right\|_2}{P_a} \leq L \frac{\varepsilon}{P_a}.$$

Therefore,

$$\mathbb{E}[u(a_{\mathrm{BR}}(f), q(f))] \geq \sum_a P_a \, u(a, \mu_a) - \sum_a P_a \cdot L \frac{\varepsilon}{P_a} = \mathbb{E}[u(a_{\mathrm{BR}}(f), f)] - mL\varepsilon.$$

Let $\hat{q} \in \arg\min_{q \in \mathcal{Q}_\varepsilon} \mathbb{E}[u(a_{\mathrm{BR}}(f), q(f))]$. Then

$$\min_{q \in \mathcal{Q}_\varepsilon} \mathbb{E}[u(a_{\mathrm{BR}}(f), q(f))] = \mathbb{E}[u(a_{\mathrm{BR}}(f), \hat{q}(f))] \geq \mathbb{E}[u(a_{\mathrm{BR}}(f), f)] - mL\varepsilon.$$

For any forecast-based policy $a(\cdot)$, optimality of the plug-in action on $f$ implies $\mathbb{E}[u(a_{\mathrm{BR}}(f), f)] \geq \mathbb{E}[u(a(f), f)]$. Moreover, since $q_{\mathrm{id}}(v) \equiv v$ is feasible for $\mathcal{Q}_\varepsilon$, we have $\min_{q \in \mathcal{Q}_\varepsilon} \mathbb{E}[u(a(f), q(f))] \leq \mathbb{E}[u(a(f), f)]$. Combining the last three displays yields the claimed inequality. $\square$

**Remark.** The proof uses only the $\varepsilon$–slack constraints for the decision indicators $\{\mathbf{1}_{R_a}\}$; any $\mathcal{H}$ that contains these tests (with per-test slack bounded by $\varepsilon$) suffices. Thus Theorem B.2 generalizes both Theorem 4.1 and Theorem 4.2.

**Bin-wise calibration under $\varepsilon$–slack: value stability and structure.** Let $\{B_j\}_{j=1}^J$ be a measurable partition of $[0, 1]^d$. Assume $\varepsilon$–*bin-wise calibration*:

$$\left\| \mathbb{E}\big[\mathbf{1}_{\{f(X) \in B_j\}} \{q(f(X)) - f(X)\}\big] \right\|_2 \leq \varepsilon, \qquad j = 1, \dots, J.$$

Write $E_j := \{f(X) \in B_j\}$, $P_j := \mathbb{P}(E_j)$, and $m_j := \mathbb{E}[f(X) \mid E_j]$ (bins with $P_j = 0$ are ignorable).

**Proposition B.3** (Value stability and piecewise-constant worst-case beliefs). *Under $\varepsilon$–bin-wise calibration,*

$$\sum_{j=1}^{J} P_j \, \text{val}(m_j) \; - \; J \, L \, \varepsilon \; \leq \; \min_{q \in \mathcal{Q}_\varepsilon} \, \mathbb{E}\big[\text{val}(q(f(X)))\big] \; \leq \; \sum_{j=1}^{J} P_j \, \text{val}(m_j).$$

*Moreover, there exists a worst-case (or arbitrarily near-worst-case) belief that is piecewise constant: for each $j$ one can take*

$$q_\varepsilon^\star(v) = p_j^\star \in \arg\min_{\|p - m_j\|_2 \leq \varepsilon/P_j} \text{val}(p), \qquad v \in B_j \text{ (a.e.)},$$

*and the robust action on $B_j$ best-responds to $p_j^\star$.*

*Proof.* For any feasible $q$,

$$\mathbb{E}\big[\text{val}(q(f))\big] = \sum_{j=1}^{J} P_j \, \mathbb{E}\big[\text{val}(q(f)) \mid E_j\big] \; \geq \; \sum_{j=1}^{J} P_j \, \text{val}\big(\mathbb{E}[q(f) \mid E_j]\big)$$

by Jensen since val is convex. The slack constraint implies

$$\big\|\mathbb{E}[q(f) - f \mid E_j]\big\|_2 = \frac{\big\|\mathbb{E}[\mathbf{1}_{E_j}(q(f) - f)]\big\|_2}{P_j} \; \leq \; \frac{\varepsilon}{P_j},$$

so, using $L$–Lipschitzness of val,

$$\text{val}\big(\mathbb{E}[q(f) \mid E_j]\big) \; \geq \; \text{val}(m_j) \; - \; L \, \frac{\varepsilon}{P_j}.$$

Summing over $j$ yields the lower bound $\mathbb{E}[\text{val}(q(f))] \geq \sum_j P_j \, \text{val}(m_j) - J \, L \, \varepsilon$. The upper bound holds because $\mathcal{Q} \subseteq \mathcal{Q}_\varepsilon$ and equality is achieved at $\varepsilon = 0$ by the exact bin-wise result.

For structure, fix any feasible $q$. Replacing $q$ by its conditional mean on each bin,

$$\tilde{q}(v) := \sum_{j=1}^{J} \mathbb{E}[q(f) \mid E_j] \, \mathbf{1}_{B_j}(v),$$

does not increase the objective (by Jensen within each bin) and preserves feasibility (the bin-wise moments are unchanged). Hence the minimization reduces to choosing, for each bin, a point $p_j \in [0,1]^d$ subject to $\|p_j - m_j\|_2 \leq \varepsilon/P_j$ to minimize $\sum_j P_j \, \text{val}(p_j)$, which yields the stated piecewise-constant form with $p_j^\star \in \arg\min_{\|p - m_j\| \leq \varepsilon/P_j} \text{val}(p)$. The best-response form of the robust action on each bin is immediate from the definition of val. $\square$

