# OpenReview forum: "Robust Decision-Making with Partially Calibrated Forecasters"
_ICLR.cc/2026/Conference — ICLR 2026 Poster_

### Official Review · Reviewer_5jx8 · 2025-10-31

**Soundness:** 4
**Presentation:** 4
**Contribution:** 3
**Rating:** 6
**Confidence:** 4

**Summary:**

The paper studies the calibration of predictive forecasts, inspired by ideas such as multicalibration.
They consider the idea of "$\mathcal{H}$-calibration", which states that a forecaster $f : \mathcal{X} \to [0,1]^d$ is calibrated if $\mathbb{E}[h(f(X)) \cdot (Y - f(X))] = 0$ for all $h \in \mathcal{H}$.
The authors analyze optimal decision-making under $\mathcal{H}$-calibration, for varying choices of $\mathcal{H}$.
A key result is that, once $\mathcal{H}$ is "rich enough" -- which they characterize by a notion of "decision calibration", weaker than the usual full calibration -- the calibration guarantee becomes already strong enough to the point of recovering the optimal decision properties of full calibration.
Optimal decision rules are also derived for some weaker choices of $\mathcal{H}$.

**Strengths:**

The paper tackles the important problem of calibration, and in particular that of achieving tractable yet powerful notions of calibration.
The result regarding the "decision calibration" choice of $\mathcal{H}$ is quite interesting, though I'm unsure whether decision calibration is truly achievable in practice.

Overall, I believe the results are a meaningful contribution to the field, from both a theoretical point-of-view (characterization of different $\mathcal{H}$s and their optimal decision rules) and a practical point-of-view (identification of certain practical notions of calibration and optimal decision rules).

**Weaknesses:**

My major concern is regarding the assumption that the utility be linear in the outcome $y$ (stated in Remark 2.1; line 210/211). It seems really strong, and I'm slightly unconvinced on how realistic it is.
Could the authors give examples besides simple expectations of (i) utilities satisfying this, and (ii) utilities *not* satisfying this? In particular, I am interested in whether there are any meaningful utilities other than simple expectations that will satisfy this.
It would also be nice to have a brief explanation on why this is necessary in the paper (as far as I can tell, this is used in the proof of Theorem 3.1 so that both Sion's minimax theorem and convexity of the adversarial problem hold; I would appreciate it if the authors could confirm or correct me). \
(It's also worth mentioning that perhaps this assumption should be in a `\begin{assumption} \end{assumption}` environment, rather than in a remark.)

I am also slightly unsure as to the actual novelty of the practical choices of $\mathcal{H}$, and in particular the general cases when (i) $\mathcal{H}$ is a finite dimensional subspace; and (ii) self-orthogonality under squared loss.
Nonetheless, I believe the paper stands on its own even if these indeed have limited novelty.
(In the case where $h$ has $X$ as the input, I am sure that these are not new, especially the finite dimensional case; but when $h$ has the forecast $f(X)$ as input I am less sure.)

Overall, this leads me to a score of 'borderline accept'. \
**Why not higher:** while I'm fairly confident with the papers contribution re. decision calibration, I am less sure of how meaningful its contributions on the weaker forms of calibration actually are. I'm also somewhat concerned with the strength of the linearity assumption on the utility. \
**Why not lower:** The paper is sound and has interesting results on a relevant problem. Its conclusions can be of broader impact, both theoretical and practical. This makes me lean towards acceptance.

These aside, I only have some minor comments:

- Line 044/045: are the observed outcomes $Y$ really in $[0, 1]^d$? I would have expected them to be in something more like $\{1, \ldots, d\}$. As far as I can tell this intended to be like a slightly more general form of "one-hot encoding" of outcomes in $\{1, \ldots, d\}$ (which would be more consistent with e.g. $E[Y | f(X) = v] = v$); am I correct?
- Lines 324-326, "Decision calibration is a minimal, task-specific threshold at which robust decision making and plug-in best-response coincide, [...]": is it really clear that there is no $\mathcal{H}$ smaller than decision calibration that still enables optimality of the plug-in best-response? I ask because of the use of "minimal" here.
- Typos / writing / formatting:
    - Line 063/064: large language models are generally neural networks, so "from neural networks to large language models" does not make much sense
    - Line 088: "informally speaking this family" -> "informally speaking, this family"
    - Line 166: "for every $\mathcal{H}$" -> "for every $h \in \mathcal{H}$"
    - Line 210: "Throughout this Paper" -> "Throughout this paper" (capitalization)
    - Line 475: broken Section `\ref`

**Questions:**

- Could the authors discuss the applicability of the linearity assumption, and elaborate on its necessity?
- Could the authors clarify the novelty of the analyses on the practical choices of $\mathcal{H}$?

---

> ### Author Response · Authors · 2025-11-20
>
> Thank you for your detailed and thoughtful review. We are glad you found the paper addresses an important problem. Your major concern regarding the linearity assumption is a crucial point, and we are grateful for the opportunity to clarify and improve the discussion around it.
>
> **On the Linearity Assumption:**
>
> Why it's necessary: You are correct that this assumption is fundamental to our approach. It is used to ensure the objective $\mathbb{E}[u(a(f(X)), q(f(X)))]$ is convex in $q$, which is required to apply Sion's minimax theorem and formulate the dual problem (as you correctly inferred from the proof of Thm 3.1).
>
> How realistic it is / Examples: The original submission (Remark 2.1) states this "captures, for example, settings in which $v$ represents a distribution over $d$ outcomes." We agree we can make this much more explicit.
>
> Consider a multi-class classification with $d$ discrete outcomes, where the decision-maker has a utility $U(a, k)$ for taking action $a$ when outcome $k$ (one of the $d$ potential outcomes) occurs. Let forecast $v$ be a probability vector $v_k = \mathbb{P}(Y_{class}=k)$. Now a natural way of defining the utility of playing an action when $v$ is the ground truth distribution of outcome is:
>
> $$u(a, v) = \mathbb{E}[U(a, Y_{class})] = \sum_{k=1}^d v_k U(a, k)$$
>
> This utility is a linear function of the forecast $v$. This gives a natural example, applicable to any multi-class classification, in which we are interested in forecasting the probability vector of outcomes. We note that risk neutral decision makers like these are the standard modeling assumption made in the vast majority of work in game theory and decision theory. Our linearity assumption captures this standard setting and is only more general. Additionally, our utility in the experiments, $u(a, y) = \alpha a y - C(a)$, is also linear in the outcome $y$.
>
>
> Non-Satisfying Example: A utility that is non-linear in the probabilities would not be covered, e.g., a risk-averse utility that depends on the variance of the outcomes.  We agree that this kind of decision maker is still interesting and we will emphasize this non-example in the revision.
>
> Action: Based on your suggestion, we have:
> - Elevated this from "Remark 2.1" to a formal "Assumption 1" to make its importance clear.
> - Expanded the discussion (as above) to provide the concrete multi-class expected utility example, showing it is a natural and common setting that fits our assumption.
> - Re-emphasized in the conclusion (as we already noted in the original submission) that extending this framework to nonlinear, risk-averse utilities is a key direction for future work.
>
> **On the Novelty of $\mathcal{H}$ Choices:**
>
> This is a fair question. We agree that properties like self-orthogonality from MSE (Prop 4.4) are known artifacts of regression. However, our contribution is not in discovering these properties, but in leveraging them as formal $\mathcal{H}$-calibration guarantees. The novelty is in connecting these "free" or "cheap" guarantees (from standard training or simple post-processing) to our minimax decision framework and showing that they can be used to derive a provably robust decision rule $a_{robust}$ that is still tractable.
>
> We also wish to clarify that our $\mathcal{H}$ classes are functions of the forecast $f(X)$ (i.e., $h: [0, 1]^d \to \mathbb{R}$), not the raw input $X$. This is crucial for a post-hoc decision-maker who may not have access to the original features $X$. We will double-check the text to ensure this is clear.
>
> **On Typos and Minor Points:**
>
> Thank you for the very careful read. We have corrected all the typos and formatting suggestions you've listed (L44/45, L324-326, L063/064, L088, L210, L475). For L44/45, we will clarify that $Y \in [0, 1]^d$ can represent, e.g., a one-hot encoding of $d$ outcomes or a $d$-dimensional regression target, and that our framework applies to both.
>
> We hope these clarifications address your concerns, and we thank you again for the feedback that has improved our paper.

---

> > ### Comment · Reviewer_5jx8 · 2025-11-25
> >
> > I'd like to thank the authors for their response. Please refer to my comments below.
> >
> > Regarding the linearity assumption:
> > that expectations satisfy the assumption is evident.
> > It's also now clear to me that these are actually the *only* utilities that satisfy the assumption (save for a normalization constant), since any utility linear in the forecast $f \in \mathbb{R}^d$ can be rewritten as $c^T f$ for some $c \in \mathbb{R}^d$, which corresponds to integration of $c$ against $f$.
> > This should be stated in the paper, as opposed to referring to expectations as only examples of the viable utilities.
> > Nevertheless, I concur with the authors that writing the utility as an expectation over the forecast is fairly common, but would like to note that non-linear utilities are of growing prominence.
> > Regarding non-linear utilities, I would have liked to see concrete examples mentioned, such as quantiles and cvar. I think it is a good idea that they be mentioned in the paper rather than a general appeal to nonlinearities.
> >
> > Now, regarding the novelty of the practical choices of $\mathcal{H}$:
> > Thank you for clarifying the contribution here.
> > I must admit I am still a bit unsure of the novelty of the optimal decision rules for these practical choices, but I'm unable to recall where I might have seen them before, so I'll let this go.
> > Just as a minor comment, please correct me if I'm wrong, but I take issue with the sentence "[The novelty is in] connecting these "free" or "cheap" guarantees (from standard training or simple post-processing) to our minimax decision framework and showing that they can be used to derive a provably robust decision rule $a_{robust}$ that is still tractable." in the rebuttal.
> > The paper derives minmax-optimal decision making rules under $\mathcal{H}$-calibration, but this hardly ensures that the decision rule is provably robust (at least insofar as robustness is taken to mean with relation to the real-world outcomes).
> > Nevertheless, I belive I understood the authors' intention with the sentence, and thank them for the clarification.
> >
> > Also, rest assured that at no point did I think that the $h \in \mathcal{H}$ were functions of $X$ directly; I only mentioned it as this is a common choice in the multicalibration literature, and which I am more familiar with. I agree with the authors that taking the forecast as input is quite reasonable.
> >
> > Overall, having pondered the submission some more, it is now clear to me that the key contribution of the submission is taking three specific choices of $\mathcal{H}$ (decision calibration, finite spans and self-orthogonality) and deriving the minimax optimal decision rule for each.
> > This is a neat contribution but of limited broader significance, in my opinion. (The result on decision calibration is still nice, though.)
> > I hence keep my borderline positive score.

---

> > > ### Author Response · Authors · 2025-11-25
> > >
> > > Thanks so much for your careful reading of the paper and your engagement with us here --- it is refreshing, and we really appreciate it.
> > >
> > > Let us say one more thing about linear utility functions, since --- although there may be a tautological sense in which all inner products <a, v> can be interpreted as expectations --- we don't think that is the most transparent way to view them when v is not in the simplex. Here are a couple of standard decision making problems that admit linear utilities that we think of as distinct from our generic statement about risk-neutral decision makers:
> > > 1) A manufacturer has a set of raw materials R which he can build a set of products P, in various combinations (i.e. with arbitrary combinatorial constraints). His utility for building a collection of products is equal to his revenue for selling the products less his cost for buying the raw materials. The unknown state of the world is the vector of prices --- both for the products in P and the raw materials in R (so a vector of length |P| + |R|). The action set corresponds to the set of feasible bundles of goods that can be manufactured, and is represented by the indicator vector for those goods minus the indicator vector for the raw materials consumed. The utility is linear in this representation.
> > >
> > > 2) In a routing problem, the set of feasible actions corresponds to s->t paths in a graph. The state of the world corresponds to a delay along each edge. The cost (or negative utility) for the decision maker is the sum of the costs along the chosen path --- so the inner product of the indicator vector of the chosen path, with the vector of costs. The utility is linear in this representation.
> > >
> > > We think of these structured examples (which can have action sets that are exponentially large in the dimension of the underlying space) as distinct from the generic example of a decision maker who has an arbitrary utility function over d outcomes, and cares about expected outcomes, where the forecast v is in the d dimensional simplex. In the above examples the distribution over the outcome space never needs to be explicitely represented, although it is true that we still require that the decision maker care about their expected outcome when it is stochastic. This is inherent for forecasts relying on calibration, which aims to represent expectations.
> > >
> > > We will make this nuance more explicit, and will point out the distinction between risk neutral decision makers and those that care about other distributional measures like CVaR and VaR.
> > >
> > > In our discussion we've also been using "robust" as synonymous with "minimax optimal" but we can also clarify.

---

### Official Review · Reviewer_ZUcV · 2025-11-01

**Soundness:** 4
**Presentation:** 3
**Contribution:** 3
**Rating:** 8
**Confidence:** 5

**Summary:**

* The paper studies decision-making with partially calibrated predictors. Particularly, the paper studies $\mathcal{H}$-calibration, a relaxation of full calibration, which also generalizes several existing calibration notions.

* It shows that best-responding to a decision-calibrated predictor is minimax-optimal, giving a simple, practical rule for decision makers.

* With weaker calibration (moment/bin-wise), it derives the tractable best-response policy.

* Empirics illustrate the robust rule's behavior vs. a naive plug-in policy.

**Strengths:**

* I really like the justification for decision calibration. The exponential sample complexity has been a problem for full calibration. The paper proves that, once satisfied, simple best response remains minimax-optimal. While full calibration is not efficiently achievable, this result is actionable for practice: decision makers need only best-respond; forecasters should focus on the concrete decision calibration objective.

* Bridges theory and empirical: connects common training artifacts (e.g., MSE first-order moments, binning) to usable robustness guarantees.

**Weaknesses:**

* The paper would benefit from a sensitivity analysis. For example, how does decision calibration loss contribute to the loss of minimax-responding decision makers? I would imagine a linear bound.

* The paper would also benefit from a discussion of the efficiency of decision calibration, given the motivation of the exponential sample complexity for achieving full calibration.


* Typos and minor issues:
  - Line 71: it's -> its
  - Line 203, citation format should be \citealp
  - Line 475, Section number missing
  - Line 132: a notable exception is Hu & Wu (2024) who do give a weaker notion than full calibration. The notion in the paper is equivalent to full calibration once the error is 0.

**Questions:**

I do not have outstanding questions, see my comments in weaknesses.

---

> ### Author Response · Authors · 2025-11-20
>
> Thank you for your positive feedback and for highlighting the practical strengths of our work. We appreciate your constructive suggestions. Below, we summarize the steps we have taken to incorporate them.
>
> **On sensitivity analysis (approximate calibration):**  In the revised version, we have added Appendix B, where we study ε-approximate $\mathcal{H}$-calibration by relaxing each calibration equality to an $\ell_2$ ball of radius $\varepsilon$. We show that the value of the minimax problem in Eq. (6), equivalently, the quantity
>
> $$
> V^{\varepsilon}:=\min _{q \in \mathcal{Q}^{\varepsilon}} \mathbb{E}[\operatorname{val}(q(f(X)))]
> $$
>
> admits two-sided bounds with at most linear degradation in $\varepsilon$. Specializing to decision calibration, Theorem B.2 proves that if $f$ is $\varepsilon$-approximately decision calibrated, then the plug-in best response is $O(mL\varepsilon)$–minimax optimal, where $m = |\mathcal{A}|$ and $L$ is the Lipschitz constant of the utility. We also give an analogous linear-in-$\varepsilon$ sensitivity result for bin-wise calibration (Proposition B.3), with worst-case beliefs that remain piecewise constant and robust actions that best-respond to slightly perturbed bin means.
>
>
>
>
> **On the efficiency of decision calibration:** This is a fair point; we can be more explicit. The sample complexity of enforcing $\mathcal{H}$-calibration is tied to the complexity of the class $\mathcal{H}$, since each test function in $\mathcal{H}$ corresponds to a constraint that requires sufficient samples to estimate. Full calibration requires a test class $\mathcal{H}$ that is infeasible to enforce with practical sample sizes (e.g., its sample complexity is exponential in $d$ [Gopalan et al., 2024]). In contrast, decision calibration’s test class $\mathcal{H_{\rm dec}} = \{1_{R_a} : a \in \mathcal{A}\}$ has cardinality $|\mathcal{A}|$, the number of actions. In many practical settings (and in our experiments, where $|\mathcal{A}| = 3$), this is a small, constant number of constraints, making it straightforward to verify. We have added a sentence to Section 4.1 clarifying that this tractability follows from $|\mathcal{H}_{\rm dec}| = |\mathcal{A}|$, which is typically small and constant, unlike the test class required for full calibration.
>
>
> **On typos and minor issues:** Thank you for the careful read. We have corrected all the typos you identified (L71, L203, L475, L132) in the revised version.

---

### Official Review · Reviewer_u2QY · 2025-11-01

**Soundness:** 4
**Presentation:** 4
**Contribution:** 3
**Rating:** 8
**Confidence:** 4

**Summary:**

Summary:
The main contribution of this paper is demonstrating how partially calibrated forecasts can be used in decision making. The authors apply a robust optimization perspective: They assume that the forecaster $f$ satisfies a weak notion of calibration called $\mathcal{H}$-calibration under the (unknown) true distribution. Then, they construct the set of distributions (which can be reduced to a set of candidate conditional expectations) under which the forecaster $f$ satisfies $\mathcal{H}$-calibration. By assumption, the true distribution must lie in this set. They propose to find the decision rule that maximizes the worst-case utility over the uncertainty set – which provides robust performance guarantees under the true distribution. They provide a characterization of the optimal robust policy under the uncertainty set. They find that when $\mathcal{H}$-calibration corresponds to notions of decision calibration, the optimal robust policy simply best responds to the forecast.

**Strengths:**

Overall, I found this paper to be clear and well-written.
To the best of my knowledge, this is the first time I have seen the connection between "partial" calibration and robust optimization -- I believe this paper proposes a novel question and framework, and also provides a clear exploration of the solution.
This is a strong contribution to the calibration literature and a very nice application of robust optimization.

**Weaknesses:**

I have questions about how the connection to this framework and decision calibration should be interpreted: To the best of my understanding, decision calibration is the “necessary” and “sufficient” condition on probabilistic forecasts required to recover the expected utility (under the true distribution), e.g. Zhao et al, 2021, Sahoo et al, 2021.
Is it the case if $\mathcal{H}$ corresponds to a set of test functions for the appropriate notion of decision calibration for our decision task, then every distribution (or conditional expectation) in the ambiguity set $\mathcal{Q}$ yields the same expected utility (which is the true expected utility)?
Furthermore, does this explain the “collapse” of the optimal minimax decision rule to the decision rule that treats the forecasts “as-is” when $\mathcal{H}$ corresponds to a decision calibration notion?
If this intuition is correct, it may be worth highlighting for readers in the main text to explain where the optimal minimax policy (for decision calibration classes) comes from.

**Questions:**

N/A

---

> ### Author Response · Authors · 2025-11-19
>
> Thank you for your positive review and for your insightful question about the interpretation of our "collapse" result (Theorem 4.1).
>
> To answer your question directly: it is **not** the case that every distribution $q$ in the ambiguity set $\mathcal{Q}$ yields the same expected utility for all policies. The set $\mathcal{Q}$ is still rich even when we have decision calibration.
>
> The "collapse" happens for a more subtle reason, which is hinted at in the proof of Theorem 4.2 (Eq. 9). The decision-calibration constraints, denoted by $\mathcal{H_{\rm dec}}$ in the paper,  are exactly the constraints required to ensure that the expected utility of the plug-in best-response policy ($a_{\rm BR}$) is invariant to the adversarial choice of $q \in \mathcal{Q}$.
>
> Specifically, for any $q \in \mathcal{Q}$ that satisfies the $\mathcal{H_{\rm dec}}$ constraints, it holds that:
>
> $$\mathbb{E}[u(a_{BR}(f(X)), q(f(X)))] = \mathbb{E}[u(a_{BR}(f(X)), f(X))]$$
>
> This means that no adversary $q \in \mathcal{Q}$ can worsen the utility of the $a_{BR}$ policy. The worst-case utility for $a_{BR}$ is identical to its nominal utility.
>
> Since $a_{BR}$ is, by definition, the optimal policy under the nominal distribution ($q(v)=v$), and our result shows that its worst-case performance is no worse than its nominal performance, it immediately follows that $a_{BR}$ must be the minimax optimal policy.
>
> We have added a short clarification in Section 4.1 emphasizing that the key mechanism behind the collapse is precisely this utility-invariance under all $q\in\mathcal{Q}$ satisfying the decision-calibration constraints. Thank you again for pointing out that this connection should be made explicit.

---

### Meta-Review · Area_Chair_RZjp · 2025-12-22

**Summary:**

The paper considers the problem of downstream decision making with respect to partially calibrated forecasts. Calibration has the property of ensuring that the optimal downstream decision rule is the one that pretends that the predicted probabilities are the ground-truth probabilities. This is a strong guarantee, but is too expensive in high-dimensional settings, which has been the subject of many recent interesting developments.

The paper shows that a relaxation, decision-calibration, provides the same guarantee in a minimax sense. This is a very nice result. Decision calibration is significantly more tractable to achieve, and the paper places it on firm conceptual standing. Therefore, the paper makes a very good contribution to better understanding relaxations of calibration.

The paper is a clear accept. It can also be considered for an oral. From this perspective, the possible downsides are that (1) the guarantee only holds in a minimax sense, so only applies for conservative decision makers. On that note, I think the discussion around lines 323 is a bit misleading, minimax optimality with respect to all decision rules is not strictly stronger than the swap regret guarantee, since the swap regret guarantee holds at a sequence-level. I think the paper should be clearer about this. (2) The paper assumes linear utilities, which is an assumption, though not such a bad one.

The reviewers had several suggestions for improvement, many of which are already included. Some additional suggestions:

1. It would be good to have some statistical significance bars for the experimental results, since the differences are small.
2. I think the paper should mention that optimality is with respect to linear decision rules in the introduction.
3. The paper should discuss the line of work on omniprediction, since it aims to provide the same guarantee of the optimal downstream
decision rule being the one corresponding to the ground-truth probabilities.

**Reviewer Concerns:**

The reviewers had several specific questions, which have been addressed.

**Reviewer Scores:**

The scores are already strong and will likely have held.

---

### Decision · Program_Chairs · 2026-01-26

Accept (Poster)